# The Wisdom of a Crowd of Brains: A Universal Brain Encoder

## Abstract

Image-to-fMRI encoding is important for both neuroscience research and practical applications. However, such "Brain-Encoders" have been typically trained per-subject and per fMRI-dataset, thus restricted to very limited training data. In this paper we propose a *Universal Brain-Encoder*, which can be trained jointly on data from many different subjects/datasets/machines. What makes this possible is our new *voxel-centric* Encoder architecture, which learns a unique "voxel-embedding" per brain-voxel. Our Encoder trains to predict the response of each brain-voxel on every image, by directly computing the *cross-attention* between the brain-voxel embedding and multi-level deep image features. This voxel-centric architecture allows the *functional role* of each brain-voxel to naturally emerge from the voxel-image cross-attention. We show the power of this approach to: (i) combine data from multiple different subjects (a "Crowd of Brains") to improve each individual brain-encoding, (ii) quick & effective Transfer-Learning across subjects, datasets, and machines (e.g., 3-Tesla, 7-Tesla), with few training examples, and (iii) we show the *potential* power of the learned voxel-embeddings to explore brain functionality (e.g., what is encoded where in the brain).

## 1 Introduction

fMRI (*functional* MRI) has emerged as a powerful tool for measuring brain activity. This enables brain scientists to explore active brain areas during various functions and behaviors (Kanwisher et al., 1997; Epstein & Kanwisher, 1998; Downing et al., 2001; Tang et al., 2017; Heeger & Ress, 2002). However, a human can spend only limited time inside an fMRI machine. This results in fMRI-datasets too small to span the huge space of brain functionality or visual stimuli (natural images). Moreover, the variability in brain structure and function responses between different people (Riddle & Purves, 1995; Frost & Goebel, 2012; Conroy et al., 2013; Zhen et al., 2015) makes it difficult to combine data across individuals that have not been exposed to the same stimuli. All of these form severe limitations on current ability to analyze brain functionality.

Image-to-fMRI encoding models, which *predict* fMRI responses to natural images, have greatly advanced the field. With the rise of deep learning, sophisticated encoding models have emerged (Yamins et al., 2014; Eickenberg et al., 2017; Wen et al., 2018c;a; Beliy et al., 2019; Gaziv et al., 2022), offering novel insights into brain function (Tang et al., 2024; Henderson et al., 2023; Gu et al., 2023). However, despite these advances, these models are primarily subject-specific and machine-specific, requiring extensive individual data (which is prohibitive) for effective training. This limits the practical use of existing brain-encoders, and prevents their ability to leverage cross-subject data. Attempts to create multi-subject encoders (e.g., (Van Uden et al., 2018; Khosla et al., 2020; Wen et al., 2018b; Gu et al., 2022)) have so far been very restrictive (see Sec. 2). These approaches have thus far not demonstrated success in merging data from multiple fMRI datasets with different stimuli and varying acquisition settings (different machine resolutions, different scanning protocols, etc.)

In this paper, we introduce the first-ever **Universal Image-to-fMRI Brain-Encoder**, which *jointly* trains on and integrates information from a collection of very *different fMRI datasets* acquired over the years (see Fig. 1). These multiple fMRI datasets provide *multiple subjects* exposed to *very different image stimuli*, scanned on *different fMRI machines* (3-Tesla, 7-Tesla), with varying number of brain-voxels (a "brain-voxel" is a tiny 3D cube of brain). What makes this possible is our new *brain-voxel centric* approach, which learns a unique "voxel-embedding" vector for each voxel of each subject. During training, each voxel-embedding learns to encode the unique visual functionality of the corresponding brain-voxel. Our Encoder trains to predict the fMRI response of each brain-voxel on any input image, by aggregating the *cross-attention* between the brain-voxel *embeddings* and multi-

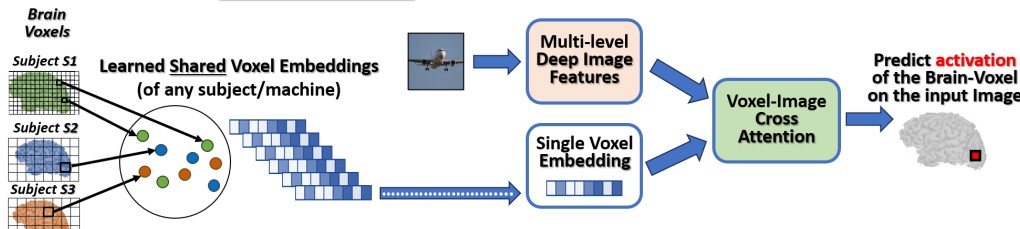

Figure 1: **Overview.** *The Universal Image-to-fMRI Brain-Encoder can train jointly on multiple subjects & datasets. It learns to predict the fMRI activation of each brain-voxel on any image, via cross-attention between learned brain-voxel embeddings and deep image features.*

level deep-features of the image. Each brain voxel (of every subject) has a corresponding voxel embedding. This embedding is merely a vector of size 256, which learns to capture what this brain-voxel is sensitive to: whether it attends to low-level image features or to high-level ones; whether it cares about the position of the feature in the image or not; etc. This voxel-embedding is initialized randomly, and is *optimized* end-to-end during training. Other than the voxel-specific embedding vectors, all other network weights are shared across all voxels of all subjects. This strategy, of learning meaningful brain-voxel embeddings via ***voxel-image cross-attention***, provides several unique benefits: (i) The *functional role* of each brain-voxel naturally emerges. (ii) The Brain-Encoder architecture is not restricted to a predetermined number of voxels per fMRI scan, a common limitation among existing brain encoders. This allows to train the encoder *jointly* on subjects scanned using fMRI machines with different scanning resolutions. (iii) When a new subject/dataset is introduced, all that needs to be learned is the new subject's voxels embeddings. Since this is captured by a small number of weights, it can be learned with few training examples.

The *per-voxel* Embedding puts a focus on individual voxel characteristics, independent of subject identity or fMRI dataset. This allows voxel functionality to be accurately captured across different subjects/datasets/machines. Moreover, the cross-attention mechanism between these voxel-embeddings and *multi-level* deep image features enables each brain-voxel to appropriately align with its corresponding "semantic level" (whether low or high).

We show the power of our approach to a variety of tasks, including: (i) Integrate information from many different fMRI datasets obtained by a "Crowd of Brains". This wealth of training data gives rise to a Universal brain encoder, whose performance/accuracy significantly exceeds that of individually-trained (subject-specific) brain-encoders. (ii) Simple *Transfer-Learning* of the Universal encoder to new subjects and new datasets, with very few training data per subject. (iii) The learned voxel-embeddings may provide a new tool to explore brain functionality, providing insights into what is encoded where in the brain.

The contributions of this paper are therefore:
- The first-ever *Universal Brain-Encoder*, which can successfully integrate data from *multiple diverse fMRI datasets* (old & new), collected on *many different subjects*, with *different fMRI machines* (3T, 7T), on very *different image datasets*.
- Universal-encoding significantly improves over individually-trained *subject-specific* encoding.
- *Transfer-Learning* of the Universal-Encoder to new subjects/datasets with few training data.
- Learn *functionally-meaningful* brain-voxel embeddings via multi-level *voxel-image cross-attention*. This powerful representation further allows to *explore brain functionality*.

## 2 RELATED WORK

**Visual Brain Encoders:** Visual brain encoders, which map complex visual stimuli to brain activity, have significantly advanced over the years. Visual brain encoders have significantly advanced the field of neuroscience by mapping complex visual stimuli to brain activity. Initially, these models utilized linear regression between hand-crafted image features, to predict fMRI responses on images (Kay et al., 2008; Naselaris et al., 2011). Over time, the field has evolved to incorporate deep learning approaches both for image feature extraction and training (Yamins et al., 2014; Eickenberg et al., 2017; Wen et al., 2018c;a; Beliy et al., 2019; Gaziv et al., 2022; Wang et al., 2023; Takagi & Nishimoto, 2023). These models are typically *subject-specific*, due to substantial differences between brain responses of different people (Riddle & Purves, 1995; Frost & Goebel, 2012; Conroy et al., 2013; Zhen et al., 2015). This not only prevents broad generalization of these models, but also restricts their training due to the limited amout of data available per subject.

**Multi-Subject fMRI Representations:** Traditionally, multi-subject fMRI studies have relied on *anatomical alignment* – i.e., canonical brain mapping to align all brains to a common anatomical space (Mazziotta et al., 2001; Talairach, 1988; Fischl, 2012; Dale et al., 1999). This often results in poor *functional* correspondences (i.e., poor alignment of multi-subject fMRI responses), due to the varied nature of brain functionality across people (Mazziotta et al., 2001; Haxby et al., 2011; Yamada et al., 2015; Brett et al., 2002; Wasserman et al., 2024). More advanced methods for *functional alignment* were recently proposed, such as Hyper-Alignment (Haxby et al., 2011; Lorbert & Ramadge, 2012; Xu et al., 2012; Haxby et al., 2020) or dimensionality reduction using auto encoders (Huang et al., 2022). However, these approaches typically require *shared data* (i.e., same images seen by multiple subjects), which significantly restricts their applicability, and cannot be used across different (mutually-exclusive) fMRI datasets. Our method overcomes these limitations by introducing a unique "voxel-embedding" for each brain-voxel, generated through predicting its response to images (via multi-level voxel-image *cross-attention*). This approach allows for tailored functional learning and information sharing across different subjects/datasets, without the need for any shared data or same fMRI machine.

**Multi-Subject Brain Encoders:** Few attempts have been made to develop models that can benefit from multi-subject brain data (Van Uden et al., 2018; Khosla et al., 2020; Wen et al., 2018b; Gu et al., 2022). None of these approaches, however, have demonstrated the capability to integrate data across multiple subjects from different datasets, with varying fMRI resolutions, or in the absence of shared data. Methods like (Van Uden et al., 2018) use Multi-Subject fMRI representations to integrate data from different subjects but, as mentioned above, these require some shared-data. Other approaches employ end-to-end multi-subject encoders with partially shared weights (Khosla et al., 2020), or use one subject's encoder parameters as the basis for another (Wen et al., 2018b), or leverage pre-trained encoder outputs for new subject adaptation (Gu et al., 2022). None of those approaches have demonstrated effectiveness on diverse datasets and machines. Moreover, they usually treat each fMRI scan as a single whole entity, thus preventing effective learning and sharing of *functional* knowledge across different brain-voxels (whether of the same person or different people). In contrast, in our *voxel-centric* Universal Brain-Encoder all network weights are shared across all brain-voxels (for all subjects/datasets/machines), *except* for the unique *voxel-specific embedding* (learned via voxel-image cross-attention). This allows our model to learn shared *voxel-functionality* across different subjects, datasets, and fMRI machines.

## 3 THE UNIVERSAL-ENCODER

Our Universal Encoder facilitates joint training on data from multiple subjects across various fMRI datasets, where subjects were exposed to completely different image stimuli and scanned using fMRI machines with differing resolutions (see Fig. 1). We first provide an overview of the method, followed by a detailed explanation of the architecture and training process.

### 3.1 OVERVIEW OF THE APPROACH

Our Universal-Encoder learns to predict the activation of each individual brain-voxel (a small cube volume within the brain) to each viewed image. A high-level overview of our Encoder's main components is provided in Fig 1 (with a detailed description in Fig 2). The model's underlying assumptions and limitations are provided in Appendix B.

The core innovation of our encoder lies in its integration of brain data with image features through a *brain-image cross-attention* mechanism. Each brain-voxel of each subject is assigned a unique

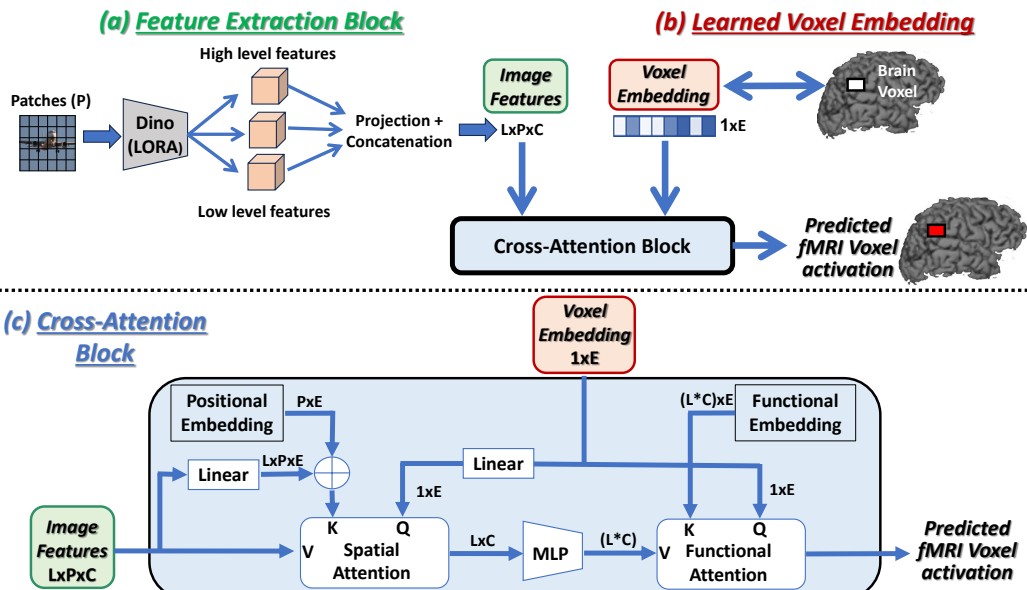

Figure 2: **Universal-Encoder Architecture.** *Input: an image & a brain-voxel index (a pointer to its Voxel-Embedding vector); Output: The predicted fMRI activation of this brain-voxel on that image. The model has 3 main components: (a) **Feature Extraction Block** – extracts multi-scale (DINO-adapted) image features; (b) **Learned Voxel-Embedding** – captures the unique functionality of each voxel; (c) **Cross-Attention Block** – establishes the connection between voxel-functionality and relevant image features*

corresponding *"voxel-embedding"* (a vector of size 256 – Fig. 2b). This embedding vector is initialized randomly and is *optimized* during training to learn to predict the fMRI response of *that* brain-voxel on *any* image (via our voxel-image cross-attention mechanism). Since this embedding is *voxel-specific* (not image-specific), yet learns to predict this voxel's activation on *any* image, it must therefore encode inside its vector the "functionality" of this specific brain-voxel (i.e., what it is sensitive to in visual data): whether it attends to low-level image features or to high-level semantic ones; whether it cares about the position of the feature within the image or not; etc.

The *shared* network components (shared across all brain-voxels of all subjects) include the ***feature extraction block*** (Fig. 2a), and the ***voxel-image cross-attention block*** (Fig. 2c). Given any image, our encoder extracts from it various image features (from low-level to high-level) using a DINO-v2 (Oquab et al., 2023) adapted model as our feature extraction block. It outputs image features from different intermediate layers of DINO, allowing each brain-voxel to attend to the appropriate semantic levels of features that align with its functionality. These image features, along with a specific voxel-embedding, are processed through the *cross-attention block*, to integrate them effectively and predict the voxel's activation in response to the given image. Note that this architecture is *indifferent to the number of brain-voxels in each fMRI scan*, hence is applicable to any fMRI data.

Our training process optimizes all 3 components simultaneously – the voxel embedding, the feature extraction block, and the cross-attention block – with the common goal of predicting the voxel response to the input image. This joint learning framework develops meaningful voxel embeddings, that not only improve voxel response prediction, but also implicitly captures its *functional role* in the brain. Our encoder and all associated weights are shared across all brain-voxels (for all subjects/datasets/machines), differing only in the per-voxel embeddings. This design ensures that each brain-voxel embedding is determined by its functional characteristics, rather than by its physical location in the brain or the subject's identity. The shared voxel embedding space supports integration of information across different voxels (whether within a single brain or across different brains). Importantly, our approach does not require subjects to have viewed the same images, nor to have been scanned in the same machine. This allows, for the first time, to integrate information from numerous fMRI datasets, collected by different groups across the globe over many years! We refer to this as the *"Wisdom of a Crowd of Brains"*.

Our proposed Universal-Encoder provides a powerful means for integrating data from multiple subjects, both within and across different fMRI datasets. These are empirically eval-

uated in Sec. 4. Moreover, it learns the *functional role* of each brain voxel, and maps *functionally-similar* brain voxels (both within the same brain, and across different brains) to nearby Voxel-Embeddings. This may potentially provide a powerful tool to explore the human brain and discover new functional regions within it. This is demonstrated in Sec. 5. What facilitates such advanced brain exploration is the enormous number of images that the large "crowd of brains" has *collectively* seen (which is prohibitive for a single subject).

## 3.2 ARCHITECTURE AND TRAINING

The architecture of our encoder is designed to receive *2 inputs*: (i) an image, (ii) a brain-voxel index (which is merely a pointer to this brain-voxel's Embedding vector), and *outputs* a single scalar value – the predicted fMRI activation of this voxel on that image. The encoder architecture comprises 3 main components: (i) the *shared* image features extraction block (Fig. 2a), (ii) the voxel embedding vectors (Fig. 2b), and (iii) the *shared* Voxel-Image cross-attention block (Fig. 2c).

**Image Features Extraction Block** (Fig. 2a). This block utilizes an adapted DINO v2 model (Oquab et al., 2023) to derive multi-scale image features. Features are extracted from L=5 intermediate layers of the DINO V2 VIT-G/14 model (layers 1,6,12,18,24), where lower layers capture low-level image features and higher layers provide more semantic information. This hierarchical feature extraction is crucial, as voxels in the visual cortex correspond to a range of image attributes – from simple visual details to complex semantic content. Each layer's features are projected to a lower-dimension C (using a linear layer), and are then concatenated along another dimension of length L (the number of layers). Given that DINO operates on P image patches, the final feature output is of size L×P×C. In order to transform Dino features into features that are suitable for predicting brain activity, we used a LoRA inspired approach (Hu et al., 2021), that is more suitable for data-limited settings (see A.2 for details).

**Per-Voxel Embedding** (Fig. 2b). Each brain voxel of each subject is assigned a voxel-specific vector of length E=256. This E-dimensional vector ("Embedding") is initialized randomly, and is optimized during training to *maximize the prediction* of this voxel's fMRI activation on different images (predicted from the cross-attention between the voxel-embedding and the image features). Since this embedding is *voxel-specific* (not image-specific), it must learn to encode inside its vector the "functionality" of this specific brain-voxel (i.e., what it is sensitive to in images). Please note that: *(i)* While each optimized embedding is *voxel-specific*, the remaining network components are *shared* by all voxels of all subjects. This allows us to train all the shared components of our Universal-Encoder on data from multiple subjects/datasets/machines, although they have varying numbers of brain voxels. *(ii)* Such joint training further facilitates the mapping of brain voxels from *different brains* of different subjects to the *same embedding space*. This allows for shared functional regions across different brains (who have never seen any shared data), to naturally surface out and be discovered. This is discussed in Sec. 5, and a few such examples are shown in Fig. 7. *(iii)* Note that unlike the common use of the term "embedding" in Deep-Learning, our voxel-embeddings are *not* an output of any sub-network. These embedding vectors are initialized randomly, and are *optimized* individually during training along with all the other *shared* components of our Universal-Encoder.

**Cross-Attention Block** (Fig. 2c). The voxel-image cross-attention block establishes the connection between voxel *functionality* and relevant visual information. This block includes three sequential components: (i) *Spatial-attention*, (ii) MLPs (multi-layer perceptrons), and (iii) *Functional-attention*. The Spatial-attention component allows the voxel embedding to select relevant locations within the image, while the Functional-attention component selects the relevant features at these locations. Both components are essential, as different brain voxels have varying image receptive fields (some voxels have small, localized receptive fields, while others are influenced by the entire image), and different functionalities (e.g., low-level versus high-level semantic features). More specifically:
*(i)* Given the features of the input image (referred to as "input features" from here on), the Spatial attention enables each voxel to focus on its corresponding spatial location within the image, effectively selecting features from the appropriate image patches. Using attention notations, the "Values" V are the input features, with dimension L×P×C. The "Query" vector $\mathbf{q}$ is the Voxel-Embedding, transformed by a linear layer which preserves its size (1×E). The "Keys" K are derived by adding a learned per-patch **positional-embedding** (size P×E) to the input features projected to the embedding size E. The output is calculated by $softmax(\mathbf{q}K_L{}^T)V_L$ for each of the L layers separately (a weighted summation across the spatial dimension P), outputting vectors of size L×C.
*(ii)* The spatially averaged features are then fed to MLPs (a separate 2-layered MLP for each of the

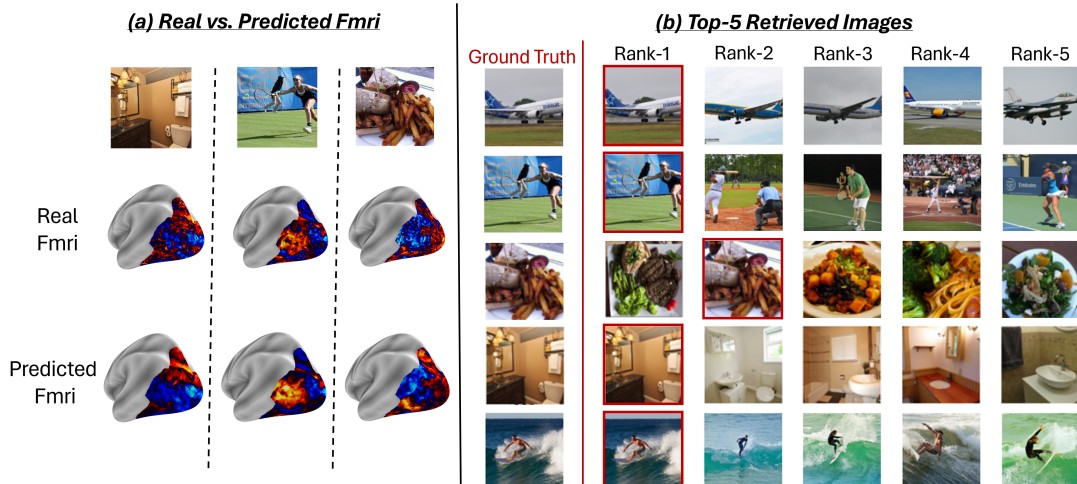

Figure 3: **Qualitative Evaluation of the Universal-Encoder.** *(a) Visual comparison of Real vs. Encoder-predicted fMRIs for 3 test images. (b) Top 5 retrieved images for each "Query" test-fMRI. (see text)*

L image-feature layers), maintaining the dimensions L×C.

*(iii)* Lastly, the Functional-attention performs a weighted summation of the spatially-attended features to derive a single scalar voxel activation. In this layer, $\mathbf{v}$ represents the flattened MLP output (size 1x(L*C) ), $\mathbf{q}$ is the voxel embedding itself, and K is learned ***functional-embedding*** that has an entry for each of the LXC features (size (L*C)XE). The output is calculated via $(\mathbf{q}K^T)\mathbf{v}^T$. This block outputs the voxel prediction as a scalar value.

**Training:** The model is trained end-to-end with all components learned together, where the objective of the model is to correctly predict the voxel activation on each input image. Our training dataset comprises images along with corresponding fMRI scans, collected on many different subjects from multiple different fMRI datasets. Training batches are constructed from 32 randomly selected images, where for each image we randomly sample 5000 voxel indices along with their corresponding fMRI activations for prediction. Each subject (brain) has its own unique voxel indices. The model is trained using the Adam optimizer with a learning rate of 1e-3. For the loss function, we employ the same loss as in (Beliy et al., 2019), using an affine combination of MSE loss and cosine proximity:

$$\mathcal{L}\left(\hat{r}, r\right) = \alpha \cdot \text{MSE}\left(\hat{r}, r\right) - (1 - \alpha)\cos\left(\angle\left(\hat{r}, r\right)\right) \tag{1}$$

where $\hat{r}$ and $r$ are the *predicted* and *measured* fMRI activations, respectively, and $\alpha = 0.1$. Training the Universal-Encoder jointly on 8 NSD subjects (see 'Datasets' below), takes ∼1 day on a single Quadro RTX 8000 GPU. Inference time (Image-to-fMRI encoding) takes ∼50 ms per-image.

## 4 EXPERIMENTS & EVALUATIONS

In this section we empirically evaluate our Universal-Encoder and its effectiveness for integrating data from multiple subjects/datasets. This section is organized as follows. We first present the datasets and the quantitative metrics used for evaluating the performance of the Universal Encoder. Next (Sec. 4.1), we demonstrate the ability of our Universal-Encoder to jointly train on multiple different subjects who were never exposed to any shared data, thus exploiting the union of all their different training sets. We further show that this exceeds the performance of any individual subject in the cohort. We then show (Sec. 4.2) that old 3T fMRI datasets with limited low-resolution data can be significantly improved by leveraging a new high-quality 7T dataset. Finally (Sec. 4.3), we show that an already trained Universal-Encoder can be easily adapted to new subjects/datasets with minimal amount of new training data, using Transfer-Learning.

**Data & Datasets:** We used publicly available fMRI datasets, which include pairs of images and their corresponding fMRI scans. These scans capture the brain activity of different subjects viewing images. A single fMRI scan provides measurements of neural activity in small volumes of the cortex (the "brain-voxels" referred to in the paper). After some pre-processing (performed by the groups who collected these data), each brain voxel is represented by a single scalar value (the *average* neural activation within that brain voxel).

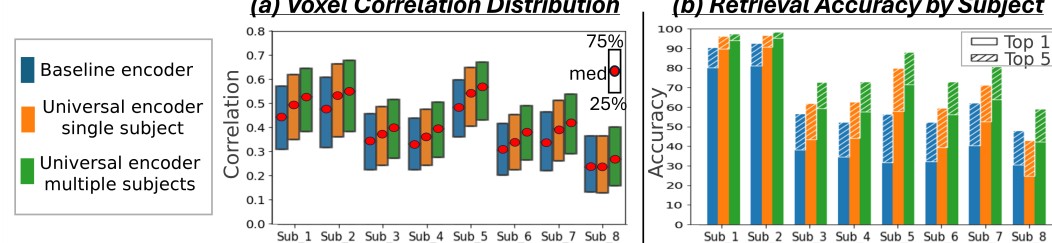

Figure 4: **The Wisdom of a Crowd of Brains.** *By aggregating data from multiple subjects, our Universal-Encoder improves the encoding of any individual subject. We compared 3 models: (i) The "Baseline" single-subject encoder of (Gaziv et al., 2022), (ii) "Universal Encoder - single subject" – our architecture trained on each subject separately, (iii) "Universal Encoder - multiple subjects" – our model trained on data from 8 subjects. (a) Pearson correlation (per voxel) between predicted & ground-truth fMRI (Median value, 75th & 25th percentiles). (b) Retrieval accuracy: ranking of the GT image within the Top-1 & Top-5 for "Query" fMRI.*

We experimented with 3 prominent fMRI datasets: (i) The old "vim-1" dataset (Kay et al., 2008; Naselaris et al., 2009; Kay et al., 2011), which features around 1750 train and 120 test *grayscale* images , and their corresponding 4-Tesla fMRI recordings for 2 subjects. (ii) The "fMRI-on-ImageNet" dataset (Horikawa & Kamitani, 2017b), which comprises 1200 train and 50 test pairs of natural images from ImageNet (Deng et al., 2009), with 3-Tesla fMRI recordings on 5 subjects. (iii) The "Natural Scenes Dataset" (NSD) (Allen et al., 2022), a new 7-Tesla dataset with 8 subjects, each having around 9000 unique *subject-specific* images, and ∼1000 images *shared* across all subjects (which we use as our test set). This results in a total of 73,000 images, all taken from the COCO dataset (Deng et al., 2009). A few example images from each of the three datasets are displayed in Figure 1. For each of these datasets, a subset of voxels (those related to visual perception) were chosen (see Appendix A.1 for more details).

**Evaluation Metrics:**   We evaluate our Universal-Encoder (i.e., its ability to correctly predict the fMRI responses of different subjects to a variety of images), using two quantitative measures. Given a set of N Test images with ground-truth fMRI per subject, we first predict the fMRI responses of those N images with our Encoder. We can then compute:

*(i) Pearson Correlation* (per voxel) – We compute the normalized correlation between the sequence of $N$ *predicted* fMRI activations of each brain-voxel (on all $N$ Test images), versus the $N$ ground-truth fMRI activations of that voxel on those images.

*(ii) Image Retrieval* (per image) – For each real fMRI scan in the test set (denoted as "Query"), we aim to retrieve (detect) its corresponding *Test-image* which produced it out of a set of N images (the Test-image and N-1 random distractors). To do so, we first predict the fMRIs of all N images in the set (using our Universal-Encoder). We then search for the Nearest-Neighbor (NN) of the *real* test-fMRI (Query) among the set of N *predicted* fMRIs (using cosine similarity). If the fMRI predicted from the Test-image is retrieved as the $1^{st}$ NN, it obtains a "Rank-1" score. If it is retrieved as the $k^{th}$ NN, it obtains a "Rank-k" score. The reported Top-1 accuracy is the percent of test-fMRIs which obtained a Rank-1 score. Top-5 accuracy is the percent of test-fMRIs which obtained a score ≤ Rank-5 (i.e., whose corresponding test image was retrieved among the first 5 NNs).

Figure 3 provides a *visual* (qualitative) example of the Retrieval-Ranking test (for Subject 1 of the NSD dataset). Fig. 3a displays the real and the Encoder-predicted fMRIs for a few sample test images. As can be seen, there is a high similarity between the real fMRI scan and the Encoder-predicted one. Fig. 3b shows the top-5 retrieved images (out of N=1000) for a few example "Query" test-fMRIs. The ground-truth test-image of each Query test-fMRI is displayed in the leftmost column of Fig. 3b. The retrieved images are ranked by the similarity of their Encoder-predicted fMRI to the real "Query" fMRI scan of the test image. As can be seen, there are many distracting (very similar) images among the 1000 test images. Yet, our Universal-Encoder is able to obtain an *average* retrieval-rank score of 1.85 (out of 1000) for Subject 1 (averaged over all 1000 Query test-fMRIs).

## 4.1 The Wisdom of a "Crowd of Brains"

We first demonstrate the Universal-Encoder's ability to exploit data from multiple subjects, without any shared-data. For this we use the new 7-Tesla NSD dataset. Our Encoder's train-set comprised the union of all the 8 subject-specific training sets (∼9000 unique images per subject), resulting in a total of ∼72,000 pairs of images with their corresponding fMRI scans. Our Encoder's test-

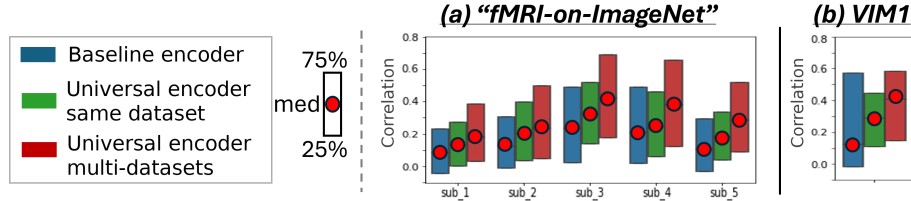

Figure 5: **The Wisdom of the Crowd of Datasets:** *Adding data from a high-quality 7-Tesla dataset (NSD) to lower-quality datasets significantly enhances encoding performance in the lower-quality datasets.*

set comprised the ~1,000 shared images (the images that all 8 subjects saw). We compared 3 models in our evaluation: (i) As a baseline, we used the image-to-fMRI encoder of (Gaziv et al., 2022), trained separately for each subject on their subject-specific training-set ("Baseline encoder"). (ii) Our Universal-Encoder trained on each subject separately ("Universal Encoder - single subject"), and (iii) Our Universal-Encoder trained on all 8 subjects jointly, using their combined training-sets, and tested on each subject individually ("Universal Encoder - multiple subjects").

Fig. 4a shows the *median* Pearson correlation value (along with the 25th & 75th percentiles, indicated by a rectangular bar around the median value), computed between the predicted activations and the ground-truth activations for all fMRI voxels. These are computed per subject, for each of the 3 models. Our Universal-Encoder, even when trained on a single subject, performs consistently better than the Baseline-encoder. Moreover, when our Universal-Encoder is trained jointly on the training sets of all subjects, it consistently outperforms all subject-specific models. It obtains notable improvements for both the "best" subjects (e.g., Subject 1 with ~7% improvement) and the "worst" subject (Subject 8 with ~15% improvement). We further evaluated the statistical significance of the performance gap between the 3 models. These are elaborated in Appendix C, and confirm the significant superiority of the Universal-Encoder. A comparison to two other prominent encoder models (Wang et al., 2023; Takagi & Nishimoto, 2023) (which have lower performance), is further provided in the appendix Fig. S8.

Fig. 4b further presents quantitative *Retrieval* results evaluated per subject for all 3 models. It shows the percent of times the correct image (corresponding to the "Query" fMRI) was ranked 1st (Top-1) and among the Top-5 retrieved images. The results indicate superior performance of the multi-subject Universal-Encoder compared to the 2 other models in both Top-1 & Top-5. Significance testing of the Universal Encoder's improvement over other models was performed using a t-test, by repeating the retrieval experiment 10 times with random image distractors and comparing the average retrieval performance for each repeat. As shown in the appendix Table. S2, the t-test reveals a maximum p-value of approximately ~4e-14. The above experiments demonstrate that our Universal-Encoder can effectively aggregate data from multiple subjects (who viewed different images), while enhancing the performance of each subject individually.

### 4.2 THE WISDOM OF A "CROWD OF DATASETS"

The Universal-Encoder can aggregate fMRI data from multiple datasets, each with own scanning resolution and unique image domain (e.g., B/W vs. color images). This allows training the Universal Encoder *jointly* on a high-quality (7T) and lower-quality (3T) datasets, thus significantly enhancing the encoding performance of old lower-quality datasets. Fig. 5 demonstrates this by training the encoder on the NSD dataset alongside two low-resolution datasets : VIM1 and "fMRI-on-ImageNet", and testing the encoding performance on individual subjects within those datasets.

Fig. 5a compares 3 models for the "fMRI-on-ImageNet" dataset: (i) The single-subject encoder of (Gaziv et al., 2022) ("*Baseline encoder*"), (ii) Our Universal-Encoder trained on all subjects in "fMRI-on-ImageNet" ("*Universal Encoder - same dataset*"), and (iii) Universal-Encoder trained on subjects from both "fMRI-on-ImageNet" and NSD ("*Universal Encoder - multi datasets*"). Our multi-subject same-dataset encoder (green) outperforms the single-subject baseline model (blue). Adding data from NSD yields even further improvement (red). Median correlation, 75th & 25th percentiles are shown. Fig. 5b shows results for the VIM1 dataset. Here too, adding training data from the high-resolution NSD dataset, significantly enhances encoding performance on VIM1.

### 4.3 TRANSFER-LEARNING TO NEW SUBJECTS/DATASETS

When a new subject/dataset is encountered, it is not necessary to train the universal-Encoder from scratch. Instead, we can add a new subject via quick Transfer-Learning, which is particularly useful

when the new subject-specific data is scarce. In our transfer-learning, all weights of the pre-trained Universal-Encoder remain fixed, and *only the 256-dimensional Voxel-Embeddings* are optimized for the new subject. This allows rapid and effective transfer learning, with little new training data.

To demonstrate the effectiveness of such Transfer-Learning, we *pre-train* the Universal-Encoder on 6 subjects from the 7-Tesla NSD dataset (Subjects 2-7). We then adapt it via transfer-learning to a new subject (without shared data) – Subject 1 in NSD, and to subjects in entirely different (older) fMRI datasets ("fMRI-on-ImageNet"(subject3) & VIM1(subject1)). Each of the 3 plots in Fig. 6 Compares: (i) Transfer-Learning of the *pre-trained* Universal-Encoder to the new subject, with varying numbers of subject-specific training data (purple curve), and (ii) a dedicated subject-specific model, trained from scratch on the subject-specific data only (orange curve). The x-axis represents the number of *subject-specific training examples*, and the y-axis shows the mean and standard deviation of the median Pearson correlation between the predicted and real fMRI scan from the new subject's test-set, over 5 runs with different random initialization & data sub-sample.

The *transferred* Universal-Encoder significantly outperforms any single-subject model on all datasets. Transfer-Learning to Subject 1 within NSD (Fig. 6a) obtains more than 77% improvement for any number of training examples. Moreover, with as little as 100 subject-specific training examples, it already achieves better results than the subject-specific model trained on the entire train-set (of 9000 training examples). A similar gap in performance is achieved for Subject 8. The transferred Universal-Encoder reaches a performance plateau at ∼1,000 examples.

Fig. 6b ("fMRI-on-ImageNet") & Fig. 6c (VIM1) demonstrate transfer learning from a *new* 7-Tesla dataset to *older* lower-resolution 3-Tesla or 4-Tesla datasets. These were scanned on very different types of images (e.g., VIM1 was scanned on B/W images with a circular black mask), and have much smaller train-sets. Transfer-Learning shows a significant improvement for any number of training data, with an improvement of more than 84% for "fMRI-on-ImageNet" and ∼45% for VIM1 when using the entire (small) subject-specific train-set. Significance testing over model initialization is shown in appendix Table. S3.

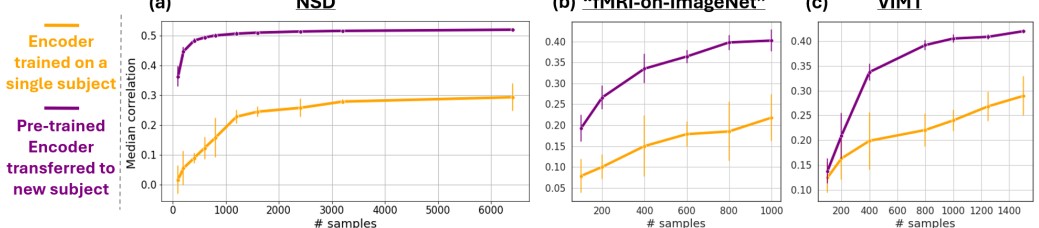

Figure 6: **Transfer-Learning of Universal-Encoder to new subjects/datasets.** *Pre-trained on NSD, the Universal-Encoder adapts to new subjects with little data. Transfer-Learning (purple) significantly outperforms single-subject models (orange). Plots show mean and standard deviation of the median Pearson correlation over 5 runs (with different random model initialization) as a function of the number of training examples.*

## 5 EXPLORING THE BRAIN USING VOXEL-EMBEDDINGS

As part of the training, our Universal-Encoder aims to learn the *functional role* of each brain voxel. It tends to map *functionally-similar* brain voxels (whether within the same brain or across different brains) to similar voxel-embeddings. This provides a potentially powerful means to explore the human brain and discover new functional regions within it. What facilitates such advanced brain exploration is the *enormous* number of images that a large "crowd of brains" has *collectively* been exposed to (which is prohibitive for any single subject). In this section, we demonstrate *initial promising findings*, suggesting that these learned voxel-embeddings capture semantically-meaningful brain functionalities, and may potentially serve as a new powerful means to explore the human brain.

While many functional parcellations of the brain exist, they all define the functional regions on a *common anatomical brain map*. These maps, although functionally-driven, are integrated across many subjects through *pure anatomical alignment*, resulting in a single functional map shared by all subjects. This results in a very coarse functional division, which overlooks individual differences. We aim to find a division that is *functionally-consistent* across different subjects, without being constrained by the need for shared anatomical mapping, thus allowing for individual anatomical (and functional) differences.

Figure 7: **Exploring The Brain:** *Clustering voxel-embeddings by their proximity in the shared embedding space allows to discover and explore functionality of brain regions. The functional role of each detected cluster of voxels is unveiled by viewing the images that most strongly activate these clusters (see text for more details).*

In order to automatically detect some shared *functional* regions across different brains, we performed the following experiment: We applied *k-means clustering* to all voxel-Embeddings of multiple "good" subjects (subjects with high prediction accuracy) from the 7-Tesla NSD dataset. Large clusters of voxel-embeddings indicate the detection of a *joined functionality*, which was learned *independently* by many different brain-voxels. Since high similarity of many independently-learned voxel-embeddings is unlikely to occur at random, we infer that what they encode is most likely *semantically-meaningful*.

To unveil the functional roles of these embedding clusters, we examine which image stimuli result in fMRI scans that induce the *highest* activation levels per cluster (averaged over all voxels within the cluster). Fig. 7a displays an example of the top 4 (out of 20) interesting discovered clusters (displayed on the brain with different colors), along with corresponding top images (images with the highest activation per cluster) for NSD Subjects 1 & 2. Each of these clusters has a clear and distinct functional role, being strongly activated by images of *Food, Faces, Text, Outdoor-Scenes*, respectively. It is interesting to note that: (i) These shared functional regions across brains surfaced out, even though the two subjects in Fig. 7a viewed *different* images; (ii) While the shared regions across the 2 subjects are in similar anatomical brain locations, they are *not anatomically aligned!* These results show that our voxel-embeddings capture functional roles rather than individual identities, hence provide a potentially powerful tool to discover shared & unique brain functionalities across different people.

We further explored the ability to detect finer functional granularity within known brain regions. Fig. 7b shows one such example – the detection of functionally-meaningful clusters (sub-regions) *within* the PPA brain region (an area corresponding to places/scenes). Two clear and distinct sub-regions have emerged on their own from our voxel-embedding clustering – *Indoor-Scenes* vs. *Outdoor-Scenes*. This demonstrates the potential power of our voxel-embeddings to uncover new functional regions beyond predefined anatomical boundaries. Additional examples & experiments are found in the appendix (Figs. S14,S15, S16).

## 6 CONCLUSION & DISCUSSION

This paper presents the first-ever *Universal* Image-to-fMRI Brain-Encoder, which can integrate data from many different subjects and different fMRI datasets collected over the years. This is facilitated by a new *voxel-centric architecture*, which learns individual "voxel-embedding" per brain-voxel, via cross-attention with hierarchical image features. In this *voxel-centric* architecture, the functional role of individual brain voxels naturally emerges, leading to better encoding performance and providing a new potential tool to explore brain functionality (e.g., what is encoded where in the brain). Moreover, this architecture is *indifferent* to the number of brain-voxels in each fMRI scan, hence can be applied to any fMRI data.

Our approach could potentially be extended to developing Brain-Encoders for other data modalities (e.g., video, audio, speech). Moreover, the Voxel-Embeddings may potentially provide a means for exploring the existence or lack of brain functionalities in 'irregular' brains (such as autistic brains, or brains of visually impaired people). Since a trained Universal-Encoder can adapt to new subjects/datasets with minimal new data via Transfer-Learning, it could potentially explore such new domains with only a small number of subject-specific scans required.

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

APPENDIX

# A    FURTHER TECHNICAL DETAILS

## A.1    FMRI DATASETS

The datasets utilized in our study comprise BOLD fMRI responses to various natural images, recorded over multiple scanning sessions. Peak BOLD responses corresponding to each stimulus were estimated, resulting in a scalar value for each voxel for each image. Each dataset underwent unique pre-processing procedures, detailed in their respective publications (Kay et al., 2008; Naselaris et al., 2009; Horikawa & Kamitani, 2017a; Allen et al., 2022).

Two additional processing steps may be needed: voxel selection from the total of all brain voxels and per voxel normalization. For each voxel, in each run, Z-scoring normalization was performed. A 'run' refers to one continuous period of fMRI scanning, and this normalization process standardizes the voxel responses across different runs, enhancing the comparability and consistency of the data.

**NSD Dataset.**    For the NSD dataset (Allen et al., 2022), we used a post-processed dataset with voxel selection from (Gifford et al., 2023). They chose vision-related areas, resulting in a total of approximately 40,000 voxels, and provided voxels after per voxel normalization.

**"fMRI-on-ImageNet" Dataset.**    For the fMRI-on-ImageNet dataset (Horikawa & Kamitani, 2017b), a relevant set of around 5,000 voxels was already provided. We implemented Z-scoring normalization ourselves for this dataset.

**VIM1 Dataset.**    For the VIM1 dataset (Kay et al., 2008; Naselaris et al., 2009), we selected the best 7,000 voxels according to the highest Signal-to-Noise Ratio (SNR). SNR is calculated as the ratio of the variance of averaged (repeated) measurements for different stimuli to the average variance of measurements for the same stimuli. This approach ensures the selection of voxels most representative of neural activity in response to diverse visual stimuli. Z-scoring normalization was also implemented for this dataset.

## A.2    MODIFIED LORA

The Lora adaptation (Hu et al., 2021) is done by adding learned low rank matrices to weights in the original network. In the original paper the value projection and query projection weights ($W_v, W_q$) in the self attention block are modified. We only modify the output projection weights($W_o$) of the self attention block.

# B    LIMITATIONS

There are two main underlying assumptions in this work that are commonly taken in most works in this area. The assumptions are that the fMRI response is memory-less and replicable. By memory-less, we mean that previous images seen by the subject do not affect the response to the current image. We wanted our model to be as general as possible and applicable to many datasets; adding memory dependence would hinder this. By replicable, we mean that the same response for an image will be measured regardless of when the subject sees the image. This is important when averaging multiple responses to the same image, a general practice in the field due to the low SNR of the fMRI signal.

Moreover, it is important to note that there is significant variability in measured signal quality across subjects. The brain exploration we demonstrate is done for subjects with relatively high SNR. For subjects with poor signal quality, it is harder to obtain a good estimation of voxel functionality, and we would likely not achieve as good results for meaningful segmentation of brain regions.

## C STATISTICAL TESTS

We conducted statistical tests to evaluate the significance of the Universal Encoder's performance across the various metrics assessed.

To evaluate the statistical significance of the prediction correlation gap between our multi-subject Universal Encoder's and the other two models as reported in 4.1 and shown in Fig. 4a , we performed a repeated measures ANOVA test. The analysis, based on the predicted voxel correlations of the 8 NSD subjects, revealed a significant performance gap between models: F-value of 72.74 and p-value of 0 (Num DF=2, Den DF=14).

Additionally, we tested the prediction correlations of individual voxels (across all subjects) against a null hypothesis. The null hypothesis was based on the distribution of correlations between pairs of independent Gaussian random vectors of size N = 982, matching the number of image-fMRI pairs in the test set. The one-sided statistical significance was estimated by comparing the predicted voxel correlations for each subject on the test set with those from the null distribution. We applied a statistical threshold of $P < 0.05$, correcting for multiple comparisons using the FDR procedure. Our results indicate that the majority of voxel predictions by the Universal Encoder are statistically significant, with an average of 98% across all subjects. Detailed results in Table S1.

|  | Subj1 | Subj2 | Subj3 | Subj4 | Subj5 | Subj6 | Subj7 | Subj8 |
|---|---|---|---|---|---|---|---|---|
| Total Voxels | 39,548 | 39,548 | 39,548 | 39,548 | 39,548 | 39,198 | 39,198 | 39,511 |
| Significantly Predicted Voxel | 39,299 | 39,024 | 38,612 | 39,059 | 39,252 | 38,665 | 38,754 | 36,850 |
| Significant Voxel Percentage | 99.37% | 98.68% | 97.63% | 98.77% | 99.25% | 98.64% | 98.87% | 93.26% |

Table S1: **Significance Test Comparison with Null Distribution.** *Number and percentage of statistically significant voxels for each subject.*

Furthermore, we present the significance of the retrieval results reported in 4.1 and shown in Fig. 4b. We performed a t-test by repeating the retrieval experiment 10 times with random image distractors and comparing the average retrieval performance per repeat of our multi-subject Universal Encoder against a competing model. The resulting p-values are presented in Table S2, where the highest p-value observed is approximately $\sim 9e-13$. This illustrates the significance of the Universal Encoder's improvements when trained on multiple subjects, compared to the other models. The results were corrected for multiple comparisons using the FDR procedure.

Lastly, we present the the significance of the transfer learning results reported in 4.3 and shown in Figure 6. We performed a t-test on the median Pearson correlation between predicted and real fMRI scans, comparing transfer learning with our universal encoder against baseline model, over 5 runs with different random model initializations, this was done for each of the dataset. Table S3 contains the estimated p-values. The results were corrected for multiple comparisons using the FDR procedure.

| Subject | Universal encoder - multiple subjects Vs Universal encoder - single subject | Universal encoder - multiple subjects Vs Baseline encoder |
|---|---|---|
| 1 | 3.795e-14 | 1.284e-23 |
| 2 | 9.275e-13 | 8.783e-16 |
| 3 | 9.848e-19 | 2.831e-25 |
| 4 | 5.274e-19 | 3.413e-22 |
| 5 | 1.407e-15 | 6.501e-22 |
| 6 | 3.413e-22 | 1.337e-23 |
| 7 | 7.104e-16 | 3.633e-21 |
| 8 | 8.491e-21 | 1.824e-21 |

Table S2: **Retrieval Results:** *Our universal encoder trained on multiple subjects significantly improves retrieval accuracy over both subject-specific models as evidenced by the calculated t-test results. t-test was conducted by repeating the experiment 10 times with random image distractors and comparing the average retrieval performance for each repeat. Table of p-values per subject illustrates the improvement significance of our universal encoder trained on multiple subjects compared to the other two models, emphasizing the significance of the improvement. The results were corrected for multiple comparisons using the FDR procedure.*

| Samples | NSD | fMRI-on-ImageNet | VIM1 |
|---|---|---|---|
| 100 | 4.85e-08 | 2.27e-05 | 1.66e-01 |
| 200 | 2.69e-06 | 4.12e-07 | 1.15e-02 |
| 400 | 1.68e-08 | 7.42e-05 | 2.56e-04 |
| 600 | 1.49e-06 | 9.32e-07 | |
| 800 | 1.87e-05 | 9.22e-05 | 4.89e-06 |
| 1000 | | 2.38e-05 | 1.39e-04 |
| 1200 | 5.18e-07 | | |
| 1250 | | | 3.97e-05 |
| 1500 | | | 1.07e-04 |
| 1600 | 3.64e-07 | | |
| 2400 | 2.69e-06 | | |
| 3200 | 2.41e-08 | | |
| 6400 | 2.01e-05 | | |

Table S3: **Transfer Learning Significance Results:** *The transferred Universal-Encoder significantly outperforms any single-subject model on all datasets. This figure shows t-test's p-values of the median Pearson correlation between the predicted and real fMRI scan for each dataset across 5 runs with different random model initialization. For each dataset we compared: (i) Transfer-Learning of the pre-trained Universal-Encoder to the new subject, with varying numbers of subject-specific training data, and (ii) a dedicated subject-specific model, trained from scratch on the subject-specific data only. This extends the findings presented in Fig. 6. The results were corrected for multiple comparisons using the FDR procedure.*

# D    ADDITIONAL FIGURES

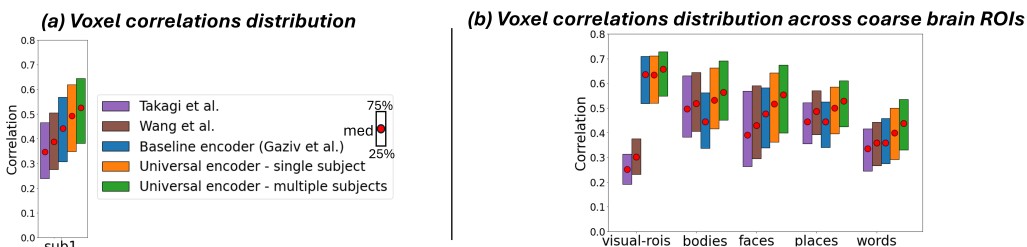

Figure S8: **Comparison against additional Encoder models.** *Pearson correlation between predicted and ground-truth fMRI of Subject1 from NSD dataset. The figure presents comparisons across three prominent encoders from different papers (Gaziv et al., 2022; Wang et al., 2023; Takagi & Nishimoto, 2023), as well as the universal encoder trained on both single-subject and multi-subject data. (a) Shows the median values along with the 25th and 75th percentiles across the five models. (b) Compares correlation scores across different coarse brain regions (ROIs) for the same five models.*

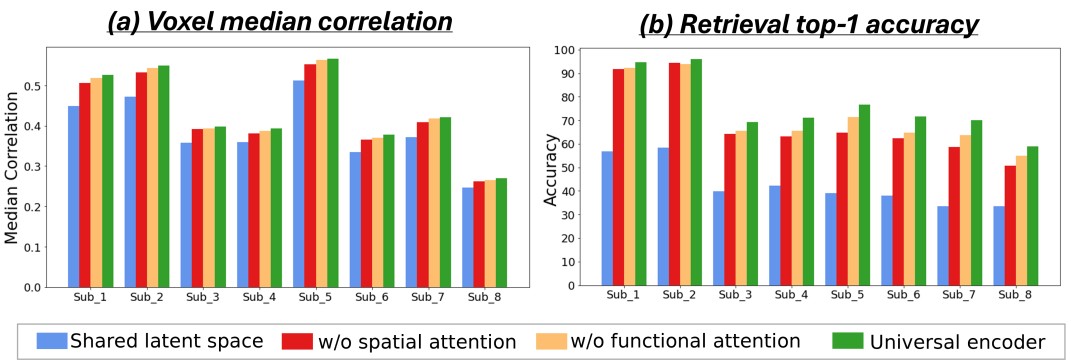

Figure S9: **Ablation of the Universal Encoder components** *We compare our full Universal Encoder model (green) to three ablated versions, all trained on data from 8 NSD subjects and evaluated on each subject's test set. (i) A shared latent space approach, where the cross-attention block is replaced with a projection to a shared latent space for all subjects (blue). (ii) Our model without the spatial attention component (red). (iii) Our model without the functional attention component (yellow). (a) Pearson correlation (per voxel) between predicted & ground-truth fMRI (Median value). (b) Retrieval accuracy: percentage of times the GT image being the Top-1 retrieved image (out of 1000) using the "Query" fMRI. As can be seen the shared latent space approach yields significantly poorer results. Furthermore, as evidenced by both the retrieval and correlation results, each component (functional and spatial) contributes substantially to performance improvement over the shared space approach, with the combination of both providing the best results.*

| Voxel Embedding Dimension | 64 | 128 | 256 | 512 | 1024 |
|---|---|---|---|---|---|
| **Median Pearson Correlation** | 0.5170 | 0.5221 | 0.5221 | 0.5227 | 0.5243 |

Table S4: **Ablation on Different Voxel Embedding Dimensions.** *Ablation study on the influence of voxel embedding dimensions E on the Universal Encoder's performance, evaluated on Subject 1 from the NSD dataset. The results show minimal sensitivity to E, indicating that encoding performance remains consistent across different embedding sizes.*

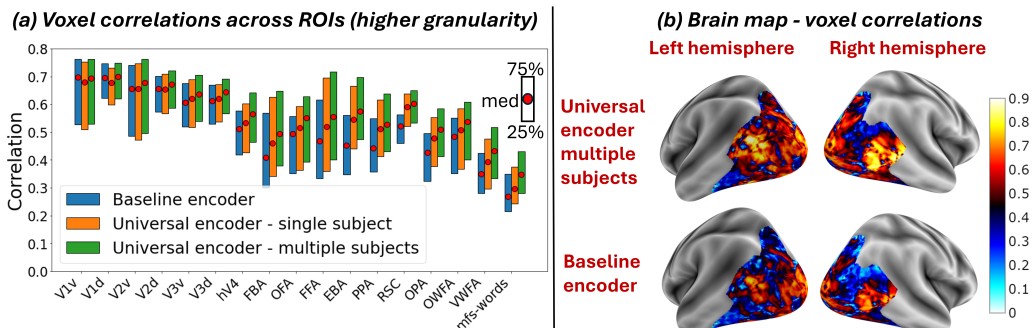

Figure S10: **Where on the brain is the error different for the different models.** *(a) Higher granularity of brain regions (ROIs), compared across the best 3 models for subject 1. (b) A visual display of the voxel correlation maps on top of the brain shown for 2 models for subject 1 – the Universal Encoder multiple-subjects versus the Baseline Encoder.*

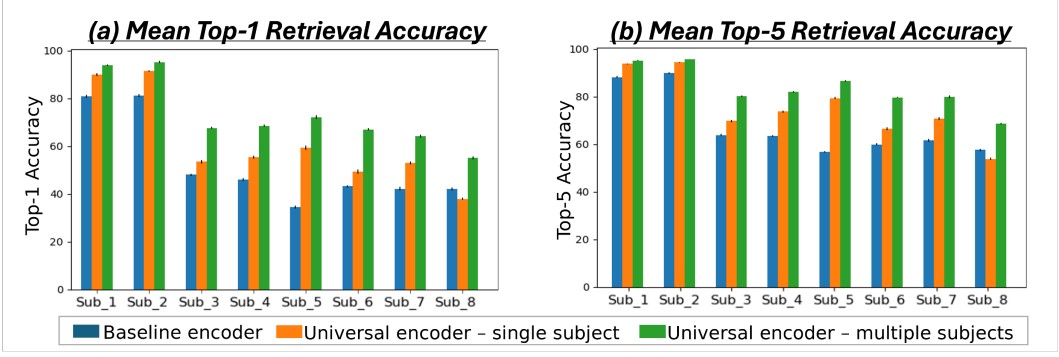

Figure S11: **Retrieval Results:** *Our universal encoder trained on multiple subjects significantly improves retrieval accuracy over both subject-specific models as evidenced by the calculated p-value. We conducted 10 retrieval experiments, wherein we randomly sampled 999 different distracted images for each "query" fMRI. Results are compared across three models: (i) "Baseline Encoder" – the encoder of (Gaziv et al., 2022) trained separately on each subject, (ii) "Universal Encoder - Single Subject" – our architecture trained separately on each subject, and (iii) "Universal Encoder - Multiple Subjects" – our model trained on combined data from 8 subjects. In (a) we present the mean top-1 accuracy (across the 10 experiments) along with the standard deviation for each model and subject. (b) depicts the mean top-5 accuracy along with the standard deviation for each model and subject.*

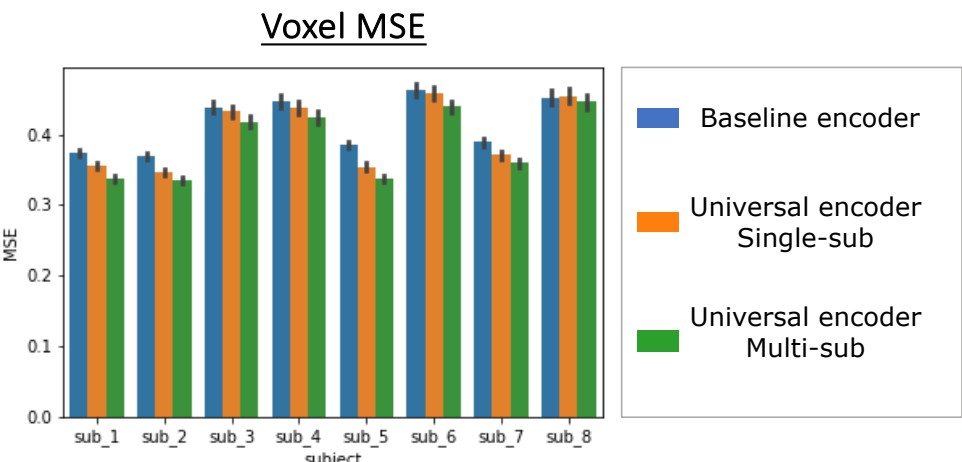

Figure S12: **MSE Evaluation - The Wisdom of a Crowd of Brains.** *By aggregating data from multiple subjects, our Universal-Encoder improves the encoding of any individual subject. We present results calculating the MSE between predicted and ground-truth fMRI (median value with 25th and 75th percentiles). We compared 3 models: (i) The "Baseline" single-subject encoder of (Gaziv et al., 2022), (ii) "Universal Encoder - single subject" – our architecture trained on each subject separately, (iii) "Universal Encoder - multiple subjects" – our model trained on data from 8 subjects.*

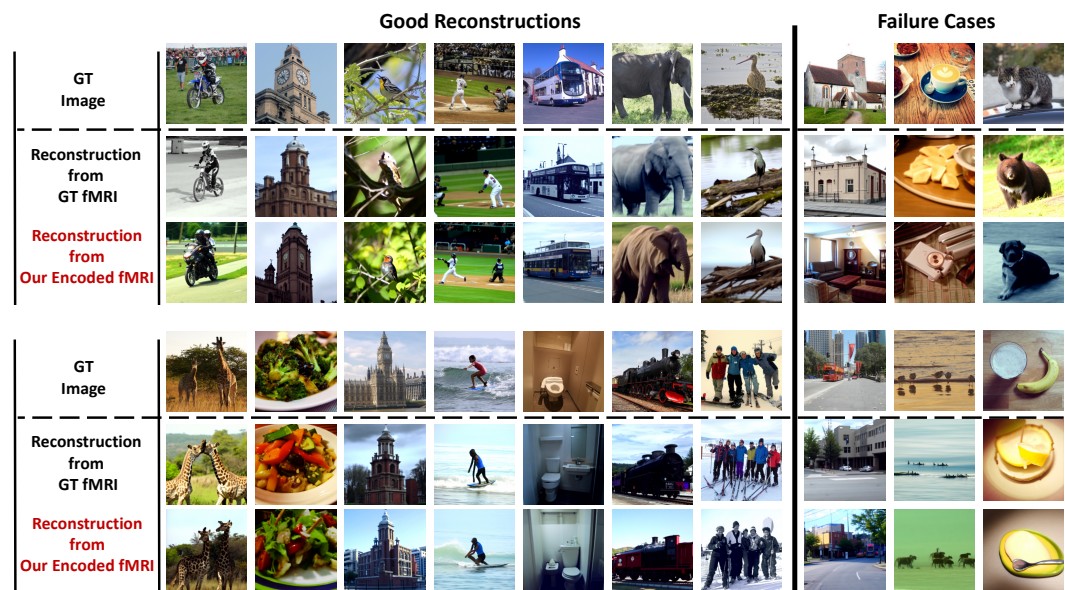

Figure S13: **Qualitative Evaluation of Image Reconstruction from fMRI.** *Visual comparison of image reconstruction from fMRI for Subject 1 from the NSD dataset. The figure showcases the ground truth (GT) image seen by the subject, alongside reconstructions generated using the fMRI-to-Image decoding model (MindEye (Scotti et al., 2024)) trained on Subject 1's fMRI data. Both the decoder and our Universal Encoder were trained on the same dataset and evaluated on the same test set. Reconstructions were evaluated on the test set using two inputs: (1) "Reconstruction from GT fMRI" where the decoder processes the real fMRI recordings of the test images, and (2) "Reconstruction from our encoded fMRI", where the test images are first encoded into predicted fMRI using our Universal Encoder before being decoded. Both successful reconstructions and failure cases are presented. As observed, the reconstructions from the encoded fMRI are on par with those from the GT fMRI, despite the decoder being trained solely on GT fMRI. This highlights the effectiveness of our Universal Encoder, demonstrating that the encoded fMRI retains information exists in the GT fMRI.*

|  | MAE ↓ | MSE ↓ | Lpips ↓ | SSIM ↑ |
|---|---|---|---|---|
| Reconstruction from GT fMRI | 4.063 | 26.965 | 0.712 | 0.139 |
| Reconstruction from our encoded fMRI | 4.089 | 27.002 | 0.710 | 0.123 |

Table S5: **Quantitative Evaluation of Image Reconstruction from fMRI.** *Quantitative evaluation of image reconstruction quality comparing the ground truth (GT) images to: (i) reconstructions from GT fMRI and (ii) reconstructions from encoded fMRI. The results presented represent the metrics averaged over approximately 1,000 images from Subject 1's test data, which serves as the test set for both the encoder and decoder. Metrics include MAE (Mean Absolute Error), MSE (Mean Squared Error), LPIPS (Learned Perceptual Image Patch Similarity, where lower values indicate closer perceptual similarity), and SSIM (Structural Similarity Index Measure, where higher values indicate greater structural similarity). Results demonstrate that reconstructions from encoded fMRI are comparable to those from GT fMRI, further validating the effectiveness of our Universal Encoder in preserving critical image-relevant fMRI features.*

**Left hemisphere**  **Right hemisphere**

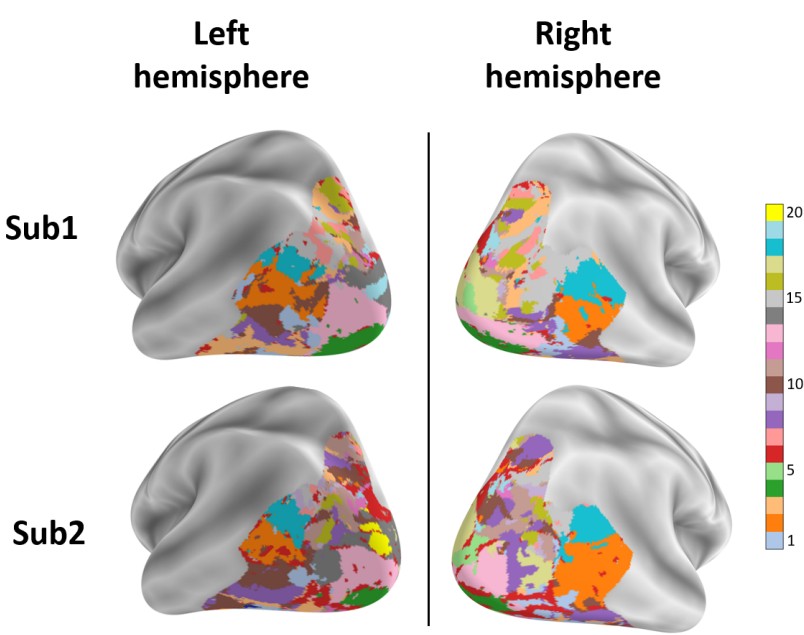

Figure S14: **Voxel Embedding Clusters:** *These are 20 clusters obtained by applying k-means to the voxel embeddings from the left and right hemispheres of subjects 1 and 2, with each cluster represented in a different color. As discussed in the paper, we further investigate the functional role of each cluster.*

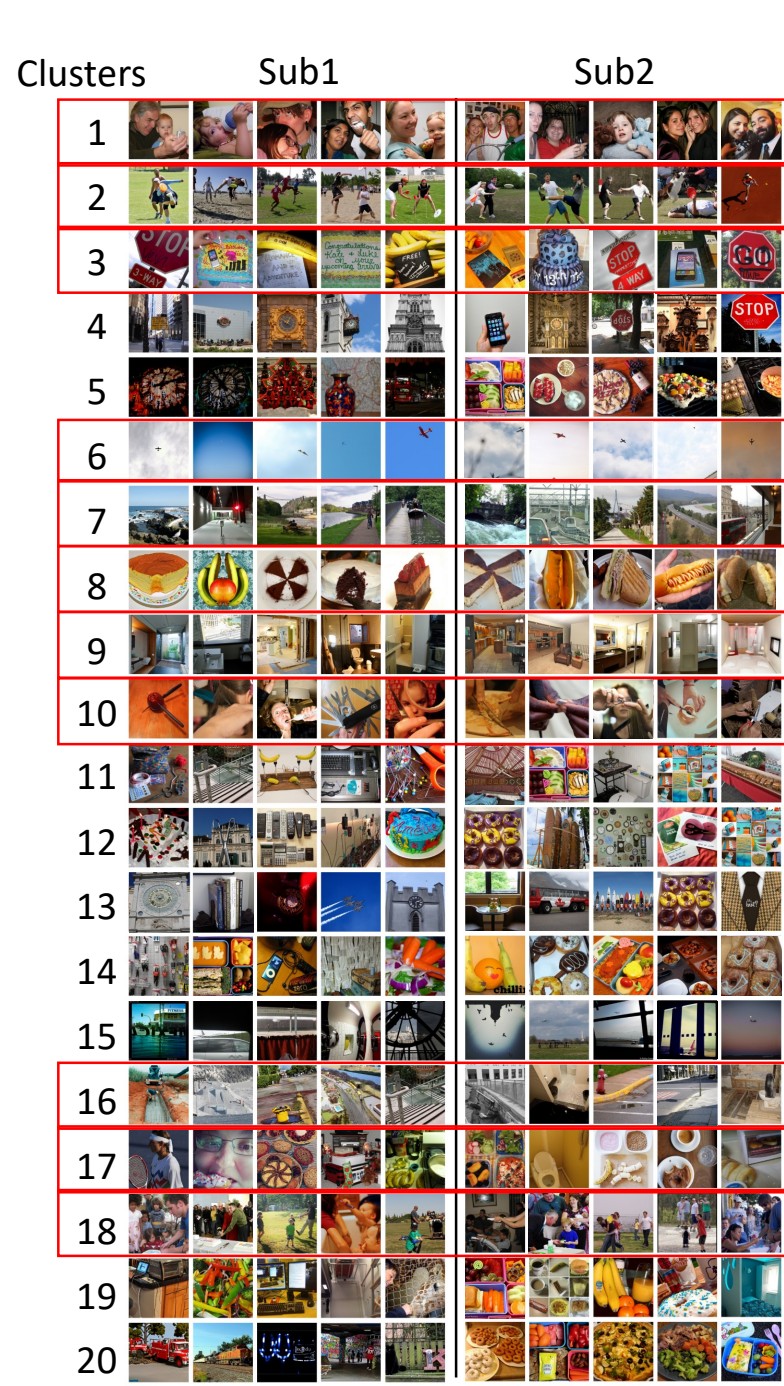

Figure S15: **Voxel Embedding Clusters:** *Top images which activated voxels within each cluster. Most clusters are consistent across subjects (marked with red frame) indicating that voxel embeddings capture functional roles rather than individual identities.*

Figure S16: **Exploring the brain:** *Clustering voxel-embeddings by their proximity in the shared embedding space allows to discover and explore functionality of brain regions. As an example, within the EBA brain region (an area corresponding to body parts), it identified functionally-meaningful clusters, revealing three distinct sub-regions. The functional role of each detected cluster of voxels is understood by viewing the images that most strongly activate these clusters, in this case: images of sports, a crowd of people and food.*

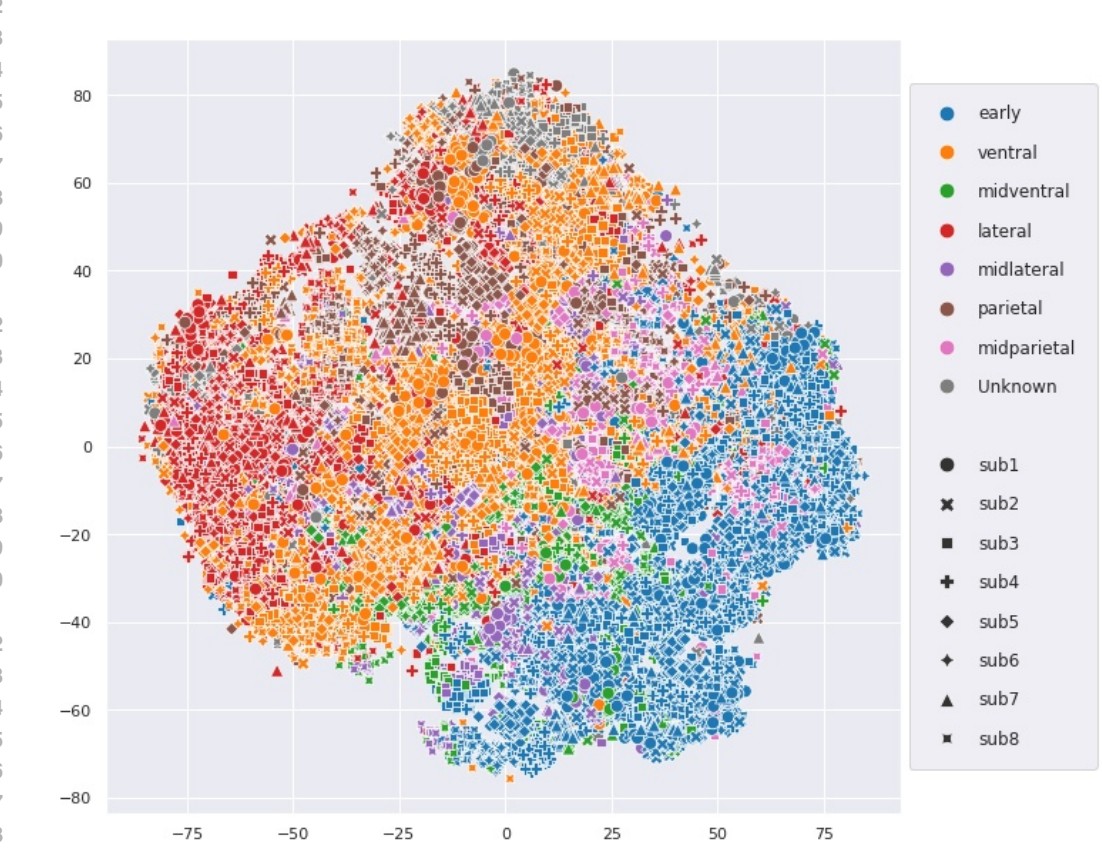

Figure S17: **2D t-SNE Visualization of Brain Streams.** This t-SNE visualization represents voxel embeddings (50,000 voxels were randomly sampled) from all 8 subjects (from NSD), with colors indicating different brain streams (e.g., early, lateral, and parietal streams) and shapes representing different subjects. The separation of brain streams in the voxel-embedding space demonstrates that our embedding effectively captures stream-specific functional properties, while embeddings from different subjects intermingle, reflecting shared functionality across subjects.

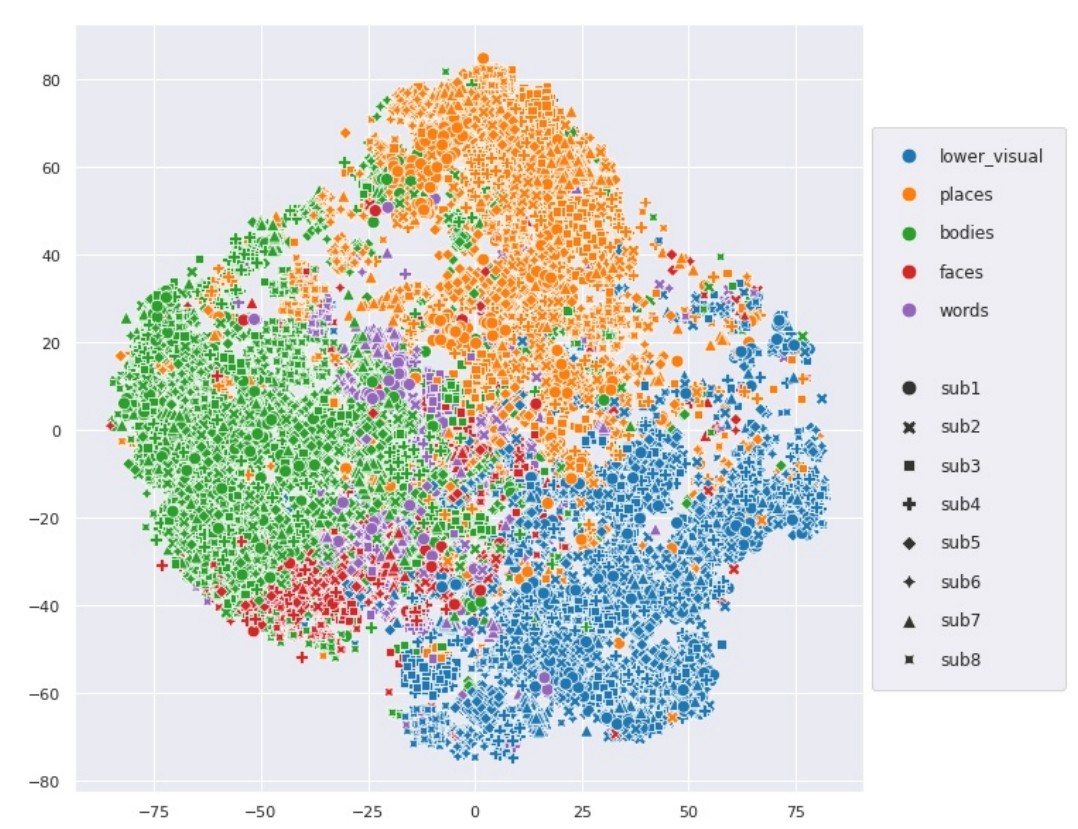

Figure S18: **2D t-SNE Visualization of Category-Selective Regions.** This t-SNE visualization highlights embeddings from four predefined category-selective brain regions: faces (e.g., FFA, OFA), bodies (e.g., EBA, FBA), places (e.g., PPA, OPA, RSC), and words (e.g., VWFA, OWFA, mfs-words). The colors signify the pre-defined cortical regions, whereas the proximity is imposed by the similarity of our voxel-embeddings. For visualization we randomly chose 50,000 voxels. As can be seen, our voxel embedding implicitly learned functionality which is similar to the functionality of known category selective brain regions.

