# OpenReview forum: "The Wisdom of a Crowd of Brains: A Universal Brain Encoder"
_ICLR.cc/2025/Conference — Submitted to ICLR 2025_

### Official Review · Reviewer_UDPX · 2024-10-30

**Soundness:** 3
**Presentation:** 4
**Contribution:** 3
**Rating:** 8
**Confidence:** 4

**Summary:**

This paper proposes a multi-subject fMRI encoding model. The authors set learnable voxel-wise embedding for each subject and optimize the subject-specific voxel-wise embedding and the subject-shared encoding model through the fMRI prediction task. Through the evaluation on multiple fMRI datasets, the authors validate the effectiveness of the proposed method and further validate that few-shot cross-subject transfer can be achieved. Finally, this paper utilizes learned voxel-wise embedding to initially explore concept-selective in the brain cortex, showing its value in neuroscience applications.

**Strengths:**

+ This paper focuses on the intersting but few-studied issue of multi-subject fMRI encoding.
+ The motivation for this paper is clear and the proposed method has yielded promising results.
+ The proposed method achieves cross-device and cross-subject few-shot transfer learning, making it highly applicable.
+ Neuroscience exploration using the voxel-wise embedding proposed in the paper is promising.
+ The paper gives detailed implementation details of the model model which helps in understanding and also ensures reproducibility.

**Weaknesses:**

#### 1. More related works should be discussed:
The authors should discuss additional related works or acknowledge that these related works have inspired them, including but not limited to:
+ At the level of model design, the proposed method seems to be a revised version of [1], which employs ROI-wise embedding rather than voxel-wise embedding.
+ At the level of research ideas, some works also train encoding models and then use them for neuroscience explorations, such as [2][3].
+ at the level of fMRI representation learning, [4][5] already show the use of multi-subject data can enhance each subject's representation, and [4][6][7] already show the use of other subject's fMRI can achieve few-shot transfer learning.

#### 2. Limited evaluation metrics:

In this paper, only voxel-wise Pearson coefficients and retrieval results are used as evaluation metrics, and the inclusion of more metrics such as $R^2$, MSE, etc. can further indicate the fMRI encoding accuracy.


#### 3. On fMRI replicable

The method proposed by the authors fails to address the issue of fMRI replicability, which is a common problem with regression-based fMRI encoding models. The authors already discuss this in their limitation and assume that the fMRI captured by subjects viewing the same image multiple times is the same. However, this assumption may greatly limit the training of fMRI encoding models.



[1] Hossein Adeli et al. Predicting brain activity using Transformers. bioRxiv, 2023: 2023.08. 02.551743.

[2] Andrew Luo et al. Brain Diffusion for Visual Exploration: Cortical Discovery using Large Scale Generative Models. NeurIPS 2023.

[3] Andrew Luo et al. BrainSCUBA: Fine-Grained Natural Language Captions of Visual Cortex Selectivity. ICLR 2024.

[4] Shizun Wang et al. A cross-subject brain decoding framework. CVPR 2024.

[5] Guangyin Bao et al. Wills Aligner: A Robust Multi-Subject Brain Representation Learner. arXiv:2404.13282.

[6] Paul S. Scotti et al. MindEye2: Shared-Subject Models Enable fMRI-To-Image With 1 Hour of Data. ICML 2024.

[7] Zixuan Gong et al. MindTuner: Cross-Subject Visual Decoding with Visual Fingerprint and Semantic Correction. arXiv:2404.12630.

**Questions:**

+ In Figure 2, the dimension of Voxel Embedding is "1xE", why 1 and not the number of voxels? If it's 1, does it mean that we need to train a model for each voxel? If it is the number of voxels, then the number of voxel embeddings should be much larger than the image tokens (i.e. P), and at this point does the attention module make the randomly initialized voxel embeddings overly self-concerned?
+ In Figure 6(a), why is the performance of coding saturated when the few-shot samples exceed 3000?

I'm willing to further raise my rating according to the author's rebuttal.

---

> ### Author Response · Authors · 2024-11-20
>
> Thank you for your detailed feedback and the time invested in reviewing our work. We have addressed your comments and will incorporate them in the updated version of the paper.
>
> **Q1.** ***The authors should discuss additional related works or acknowledge that these related works have inspired them.***
>
> **Response:** Thank you for the references. Below we address the papers/approaches you referred to. We will also incorporate this discussion into the revised version (in the Prior Work section). More specifically:
> - While our model design bears some resemblance to [Adeli et al], there are fundamental differences. Unlike ROI-wise learned embeddings, our voxel-wise embedding approach enables integration of data across different brain voxels, both within and across different ROIs (and also within & across brains). Moreover, our voxel-based approach does not require any prior ROI parcellation of the brain.  Finally, as can be seen in Fig. 7 in our paper, functional parcellation can be different across different subject brains (even though the brains are anatomically aligned). We will clarify those important differences.
> - Regarding the potential uses of Image-to-fMRI encoding for neuroscience exploration (e.g., [Luo et al – 2023,2024]) – Thank you for pointing out these interesting papers. We will definitely add these references to our Introduction, to further motivate the importance of developing Image-to-fMRI encoding. Indeed, our new voxel-centric approach facilitates additional neuroscience exploration tools by leveraging data from multiple subjects and datasets.
> - Regarding prior work on fMRI representation learning (refs [4-7] suggested by the reviewer) – First, these papers do not address the problem of Image-to-fMRI encoding (which is the focus of our paper). Secondly, these methods are all fMRI-centric. Namely, they treat each fMRI scan as a single entity, relying on shared embeddings for the entire fMRI across subjects. In contrast, our model is VOXEL-centric, hence searches for shared VOXEL-embeddings across & within subjects. Our approach shares network weights across all brain-voxels (irrespective of subject, dataset, or machine), with only the voxel-specific embeddings being unique. Similarity between different voxels within & across brains is more likely to exist, than similarity between entire fMRI responses across different brains. This fundamental difference is what allows: (i) Effective integration of data across different subjects/datasets/machines.(ii) Simple transfer learning of Image-to-fMRI encoding to new brains/datasets with few data samples. (iii) Facilitates exploration of brain regions/functionality.
>
> Despite the above-mentioned differences, we agree that these papers are closely related, and will refer to them properly in our Prior Work section.
>
> **Q2.** ***Limited evaluation metrics: In this paper, only voxel-wise Pearson coefficients and retrieval results are used as evaluation metrics, and the inclusion of more metrics such as R2, MSE, etc. can further indicate the fMRI encoding accuracy.***
>
> **Response:** We have now also added MSE results for the experiments in Figure 4a (see Fig. S11 in the revised appendix). We have also added another type of evaluation of the encoded fMRI quality via image-reconstruction from fMRI (using a pre-trained fMRI Decoding model) – see Fig. S12 & Table S5 in the revised appendix.
>
> **Q3.** ***On fMRI replicable: The method proposed by the authors fails to address the issue of fMRI replicability, which is a common problem with regression-based fMRI encoding models. The authors already discuss this in their limitation and assume that the fMRI captured by subjects viewing the same image multiple times is the same. However, this assumption may greatly limit the training of fMRI encoding models.***
>
> **Response:** We agree that our method does not address the issue of fMRI replicability, which remains a limitation of our work (as mentioned in our paper). However, this challenge is not unique to our approach; it is a common issue faced by most existing Image-to-fMRI encoding models, which similarly assume consistent fMRI responses across repeated stimuli presentations.

---

> ### Author Response · Authors · 2024-11-20
>
> **Q4.** ***In Figure 2, the dimension of Voxel Embedding is "1xE", why 1 and not the number of voxels? If it's 1, does it mean that we need to train a model for each voxel? If it is the number of voxels, then the number of voxel embeddings should be much larger than the image tokens (i.e. P), and at this point does the attention module make the randomly initialized voxel embeddings overly self-concerned?***
>
> **Response:** Indeed, in a single training batch we estimate/train the voxel embeddings for multiple voxels simultaneously (5,000 randomly sampled voxels per image in each training batch, with 32 images per batch – see Line 297-299 in the paper). Fig.2 illustrates the process applied to each individual voxel-embedding (hence  "1xE").  All the other components of the encoder (except the voxel embedding) are shared by all voxels.
>
> Note that each single voxel embedding must learn to predict the activation of a specific voxel on ALL input images, ensuring it cannot be overly self-referential. While the number of tokens per image (P=169) is much smaller than the number of voxel embeddings (~300,000), each voxel embedding must learn to predict its own activations on all images in its training data. Since each subject (voxel) saw ~9,000 subject-specific images,  the total number of image tokens this voxel is exposed to is 9000xP = ~1,500,000 image tokens.
>
> **Q5.** ***In Figure 6(a), why is the performance of coding saturated when the few-shot samples exceed 3000?***
>
> **Response:** The incline of the graphs beyond 3000 examples is indeed slow, although not fully saturated. A possible explanation is the diminishing return phenomenon, where the rate of gain decreases as the dataset size grows. Initially, each new sample adds substantial information, but as the dataset increases, additional samples offer less. However, other factors specific to voxel quality and functional variability may also play a role in this behaviour.

---

> > ### Comment · Reviewer_UDPX · 2024-11-20
> >
> > Dear authors,
> >
> > Thanks for your reply, I have carefully read through your feedback. Your reply has solved all my doubts.
> >
> > I am familiar with some works in the field of fMRI encoding. In my opinion, this manuscript is good. On the one hand, the paper realizes multi-subject, cross-device fMRI encoding and the few-shot encoding; on the other hand, the proposed learnable voxel-embedding is helpful for neuroscience research. In addition, the manuscript is refreshingly written.
> >
> > In summary, this paper has sufficient contribution to the field of fMRI encoding and should be accepted by ICLR.
> >
> > Reviewer UDPX

---

> > > ### Comment · Reviewer_UDPX · 2024-11-20
> > >
> > > I will revise my rating (from 6 to 8) . Look forward to the code and model weights going open source. Good luck！

---

### Official Review · Reviewer_BQD5 · 2024-11-03

**Soundness:** 2
**Presentation:** 3
**Contribution:** 3
**Rating:** 6
**Confidence:** 4

**Summary:**

This work proposed a Universal Brain Encoder that can train on multiple subjects from different datasets. The methods applied cross-attention between features of the image and fMRI voxel. Experiments on three datasets have shown that performance can be improved after fine-tuning with new subjects.

**Strengths:**

1. The problem of the universal brain-image generation is important.

2. Experiments successfully demonstrated the proposed method can train on multiple subjects from different datasets and achieved a better performance.

3. The presentation of motivations, methods, and experiments is clear and easy to follow.

**Weaknesses:**

1. The announcement of the *first-ever Universal Brain-Encoder* is too aggressive. The idea of the model is to be able to train on multiple subjects and datasets instead of **universally** applying to any unseen subjects or datasets. The performance of the proposed model on a new subject is tested via few-shot transfer learning instead of zero-shot learning.

2. The method of cross-attention is not novel and exists in the field of brain-image generation [1,2,3].

[1] Sun, Jingyuan, et al. "Contrast, attend and diffuse to decode high-resolution images from brain activities." Advances in Neural Information Processing Systems 36 (2024).

[2] Wang, Shizun, et al. "Mindbridge: A cross-subject brain decoding framework." Proceedings of the IEEE/CVF Conference on Computer Vision and Pattern Recognition. 2024.

[3] Scotti, Paul S., et al. "MindEye2: Shared-Subject Models Enable fMRI-To-Image With 1 Hour of Data." arXiv preprint arXiv:2403.11207 (2024).

3. Lack of ablation studies. For example, in Figure 7, functional embedding is not evaluated to show the functional roles of voxels but the k-means results of voxel embeddings were used. Also, other components, which were claimed essential in the main text, are not evaluated.

4. The potential power of finding subject-specific brain parcellation is interesting, but the demonstration in Figure 7 shows this can only proceed on visual networks instead of the whole brain. Brain parcellation is for the whole brain.

**Questions:**

See weaknesses.

---

> ### Author Response · Authors · 2024-11-20
>
> Thank you for your time and comments. Below we address each of your concerns.Please let us know if this resolves your concerns, or whether there are additional comments/requests.
>
>
> **Q1.** ***The announcement of the first-ever Universal Brain-Encoder is too aggressive. The idea of the model is to be able to train on multiple subjects and datasets instead of universally applying to any unseen subjects or datasets. The performance of the proposed model on a new subject is tested via few-shot transfer learning instead of zero-shot learning.***
>
> **Response:** In the context of our work, the term "Universal" refers to the capability of our Brain-Encoder to train on multiple subjects and datasets simultaneously, although they share no data. In contrast, most previous approaches train for each subject its own independent subject-specific Brain-Encoder. This will be further clarified in the paper.  Training a high-quality Image-to-fMRI encoder in a zero-shot setting is inherently infeasible because subject-specific brain functionality must be learned (as anatomical alignment yields poor functional prediction). However, as demonstrated in the paper, our approach also allows for simple and high-quality transfer learning to new subjects and datasets, with very few examples (i.e., few-shot transfer learning).
>
> **Q2.** ***The method of cross-attention is not novel and exists in the field of brain-image generation [1,2,3].***
>
> **Response:** Indeed, cross-attention has been used before (and is not our claim). What we present, for the first time, is the ability to perform such cross-attention at a VOXEL level. This is what enables our model to handle data from multiple subjects with different numbers of brain-voxels, thereby allowing our model to integrate the data across different datasets. We believe this specific mechanism of cross-attention for image-to-fMRI encoding is a unique contribution of our work.
>
> **Q3.** ***Lack of ablation studies. For example, in Figure 7, functional embedding is not evaluated to show the functional roles of voxels but the k-means results of voxel embeddings were used. Also, other components, which were claimed essential in the main text, are not evaluated.***
>
> **Response:** We have included several ablation studies in the paper and in the Supplementary-Material. Specifically, we provided an ablation study separating the contributions of the new architecture from the impact of data integration across subjects and datasets (Figures 4 and 5). Additionally, voxel embeddings were quantitatively evaluated by comparing models trained on a single subject versus multiple subjects (Figures 4, 5 and S8).
>
> Regarding Fig.7 – individual voxel measurements can be inherently noisy, which is why we preferred to examine clusters of voxels to evaluate their functionality. This aggregation offers a more robust analysis, and provides more meaningful visualizations. Note that visualizing each individual voxel for all subjects would be impractical due to the sheer number of voxels involved. If the reviewer thinks any specific additional evaluations are needed, we are open to adding them if feasible.
>
>
> **Q4.** ***The potential power of finding subject-specific brain parcellation is interesting, but the demonstration in Figure 7 shows this can only proceed on visual networks instead of the whole brain. Brain parcellation is for the whole brain.***
>
> **Response:** We agree with the reviewer that subject-specific brain parcellation for the entire brain would be wonderful and would yield broad insights.The current stimuli that the subjects were exposed to was visual, therefore the parcellation is of visual brain areas. For audio  stimuli, for example, a similar parcellation of audio brain areas could be achieved by our approach (with the proper adaptation of the method). Indeed, this is part of our planned future work.

---

> > ### Comment · Reviewer_BQD5 · 2024-11-20
> >
> > Q1: My major concern of this paper to be accepted is the soundness and contribution of the proposed method. It is undoubtable the performance will be enhanced by using more training data from multiple subjects, but what rooted the problem of subject-dedicated training and what makes the proposed method can address the problem? Does the problem here similar to the challenges of cross-subject brain-decoder, e.g., (1) size variability and (2) diverse neural response by MindBridge?
> >
> >  - Wang, Shizun, et al. "Mindbridge: A cross-subject brain decoding framework." Proceedings of the IEEE/CVF Conference on Computer Vision and Pattern Recognition. 2024.
> >
> > Q3 (Fig 7): I was confused by using voxel embedding to indicate the functional role. Doesn't your functional embedding is the one to play the role of bringing functional role of voxels? Like the positional encoding in Transformers. How the input of your model (the cross-attention block) indicating the functional role?
> >
> > Q3 (Ablation study): In addition to using multiple subject as training data, which is undoubtable can enhance the accuracy, you should add ablation studies by dropping some components of your model, e.g., positional/functional embeddings in the cross-attention block, to indicate the proposed method is contributing to the performance improvement. I can't find these results.

---

> ### Author Response · Authors · 2024-11-21
>
> **Q1.** ***My major concern of this paper to be accepted is the soundness and contribution of the proposed method. It is undoubtable the performance will be enhanced by using more training data from multiple subjects, but what rooted the problem of subject-dedicated training and what makes the proposed method can address the problem? Does the problem here similar to the challenges of cross-subject brain-decoder, e.g., (1) size variability and (2) diverse neural response by MindBridge?***
>
> **Response**: Efficiently integrating data from multiple subjects is not trivial, thus getting significant performance improvement from multiple subjects is an inherent challenge. Indeed, as noted by the reviewer, the main problem in multi-subject training (also noted in the MindBridge paper) is the diverse neural response across subjects.
>
> To handle this diversity across subjects, “MindBridge” and other papers developed subject-specific (fully-connected) networks that map each subject’s fMRI to a shared latent-space, which captures the common denominator across all subjects. This is a useful approach for fMRI-to-Image DECODING, since the output of the decoder (an image) is also in a shared space across all subjects (i.e., the image space). In contrast, in Image-to-fMRI ENCODING, the input (an image) resides in a shared space for all subjects, whereas the outputs (fMRIs) are different and diverse across subjects. Thus, relying solely on projecting data into a shared space is inherently problematic, as this would not preserve subject-specific brain functionality. To address this inherent problem, we propose a VOXEL-centric approach, where per-voxel embeddings capture subject-specific functionality, while still leveraging shared patterns across subjects via our cross-attention network components (which are shared across all hundreds of thousands of VOXELS, and not only across subjects).
>
> Please note that all previous methods (both encoding and decoding) are predominantly fMRI-centric. Namely, they treat each fMRI scan as a single entity, relying on shared embeddings for the entire fMRI across subjects. In contrast, our VOXEL-centric model shares network weights across all brain-voxels (both within and across subjects, irrespective of subject identity, dataset, or machine), with only the voxel-specific embeddings being unique. Similarity between different voxels within & across brains is more likely to exist, than similarity between entire fMRI responses across different brains. This fundamental difference is what allows: (i) Effective integration of data across different subjects/datasets/machines.(ii) Simple transfer learning of Image-to-fMRI encoding to new brains/datasets with few data samples. (iii) Facilitates exploration of brain regions/functionality.
>
>  We will clarify those differences in the paper. Please let us know if this answers your question.
>
> **Q3a.** ***The role of functional-embedding & voxel-embedding.***
>
> **Response**: We believe we previously misunderstood your question in our earlier response, and apologize for that. The functional embedding indeed plays an essential role in enabling each voxel embedding to attend to relevant image features and determine what in the image is significant. The functional-embedding is a learned matrix that interacts with each voxel-embedding allowing it to attend to its relevant image features. It is identical for all voxels, and therefore does not carry voxel-specific information. The voxel-embedding, on the other hand, is responsible for learning the unique functionality of each voxel. Since the voxel-embedding is voxel-specific (not image-specific), yet it learns to predict this voxel’s activation on any image, it must therefore encode inside its vector the “functionality” of this specific brain-voxel (i.e., what it is sensitive to in visual data): whether it attends to low-level image features or to high-level semantic ones; whether it cares about the position of the feature within the image, or not. Therefore, in Fig. 7, when exploring the brain to uncover voxel functionality, we rely on the voxel-embeddings.
>
> **Q3b.** ***Ablation studies.***
>
> **Response**: Per the reviewer’s request, we will conduct ablation experiments to assess the impact of removing the positional and functional embeddings. We are currently working on setting these ablation experiments; it will take a few days to get all the results.
>
> Having said that, as explained above, our new voxel-centric architecture is what facilitates effective cross-subject learning. Consequently, our evaluation focused on demonstrating the effectiveness of this integration within and across datasets.

---

> > ### Comment · Reviewer_BQD5 · 2024-11-21
> >
> > Q1: Does the voxel-centric refer to a single voxel as the input of the model? If so I totally missed this point as a methodology contribution, which should be highlighted as the main difference between previous methods. The term "voxel-centric" is rarely present in your manuscript and the bolded "cross-attention" in the Introduction and figures misled me. I will raise the contribution score accordingly.
> >
> > Q3b: I would still ask for these two ablation studies. I think these are important to indicate how different components in your model contribute the performance improvement. As displayed in Fig S8, you should illustrate the main results along with ablation studies and some existing methods instead of solely with the basic/ablation version of your own model. This can be the empirical evidence to support your contributions. I will raise the soundness score if the results of ablation studies fit with your methodology philosophy.

---

> > > ### Author Response · Authors · 2024-11-23
> > >
> > > Thank you again for your time and effort, both in your initial review and during this discussion period.
> > >
> > > ***Q1 Response:*** Yes indeed, our voxel-centric approach takes as input an image and a single brain-voxel index (a pointer to its Voxel-Embedding vector), and outputs the predicted fMRI activation of this brain-voxel on that image. We will clarify and emphasize the role of our voxel-centric approach in the revised paper.
> > >
> > > ***Q3b Response:*** We have now added to the revised appendix the reviewer-suggested ablation studies, along with an additional one that further demonstrates our model’s contributions. In Fig.S9, we compare our full Universal Encoder model to three ablated versions, all trained on data from 8 NSD subjects and evaluated on each subject’s test set:
> > >
> > > (i) Our model without the spatial attention component.
> > >
> > > (ii) Our model without the functional attention component.
> > >
> > > (iii) A shared latent space approach. The cross attention block is replaced with projection to shared latent space for all subjects (Similar to fMRI-centric methods).
> > >
> > > As shown in the figure, the shared latent space approach yields significantly poorer results, likely due to its inability to retain subject-specific information and its limited capacity to effectively share information across voxels. Furthermore, as evidenced by the retrieval and correlation results, each of the components (functional and spatial) provide a substantial improvement over the shared space approach. While together they provide the best results. We will address this in the final version of the paper.

---

> > > > ### Comment · Reviewer_BQD5 · 2024-11-23
> > > >
> > > > Thanks for your effort. It looks like spatial attention leads to the highest improvement for the most of subjects. It would be better if you could add a brief discussion of these results in the main text.
> > > >
> > > > The authors have addressed all my concerns. I'd like to raise my score to 6.

---

### Official Review · Reviewer_r8MQ · 2024-11-04

**Soundness:** 4
**Presentation:** 4
**Contribution:** 4
**Rating:** 8
**Confidence:** 3

**Summary:**

This work trains a universal brain encoder that can predict the brain responses from multiple participants and datasets. The input to the encoder is a stimulus image and a learned 256-dimensional voxel-wise embedding. The voxel-wise embeddings are randomly initialized, and contain no information about the participant or the spatial location of the voxel. All other parameters in the encoder are shared between voxels, participants, and datasets.

- The predictions are evaluated with per-voxel pearson correlation and a per-image retrieval metric.
- Their universal encoder significantly outperforms baseline single-subject encoders (figure 4)
- Inclusion of a higher quality 7T data improves performance on older 3T and 4T datasets (figure 5)
- A pre-trained encoder can transfer to new subjects and datasets just by learning new voxel embeddings. Performance is much higher and learning is faster than a single-subject encoder.
- K-means clustering is applied to the voxel embeddings to identify regions for food, words, faces, sports, indoor scenes, outdoor scenes.

**Strengths:**

- This is a very strong and well written paper. The methods are easy to understand and well motivated. I could see this encoder being used a lot when working with smaller vision datasets.
- Retrieval accuracy is impressively high. It looks close to 95% top-1 accuracy for subjects 1 and 2 across 1000 test images (chance is 0.1%).
- Statistical tests are performed for all experiments.

**Weaknesses:**

I think the paper is lacking some exploration and visualization of the voxel embeddings. Here are some ideas:
- Apply the clustering to more than 2 participants.
- Other clustering methods besides k-means (i.e. some that can deal with outliers)
- A flatmap visualization with outlines of previously identified category selective regions for faces, bodies, places, and words. This would be helpful for comparing to the clusters identified with k-means.
- A UMAP or tsne applied to the combined embeddings for the 8 participants, and then visualized on the cortical surface with a color mapping.

**Questions:**

In other papers that use NSD, subject 5 is typically the best performing subject. However in this work the retrieval accuracy is quite a bit lower than subjects 1 and 2. Any ideas why this might be the case?

---

> ### Author Response · Authors · 2024-11-20
>
> Thank you for your time and effort, and for the suggestions. We have done our best to address them and include the additional suggested visualisations in the revised appendix.
>
>
> **Q1.** ***Lacking some more exploration and visualization of the voxel embeddings. Here are some ideas: (a) clustering to more than 2 participants; (b) Other clustering methods besides k-means (i.e. some that can deal with outliers); (c) A flatmap visualization with outlines of previously identified category selective regions for faces, bodies, places, and words; (d) A UMAP or tsne applied to the combined embeddings for the 8 participants, and then visualized on the cortical surface with a color mapping.***
>
> **Response:** We have tried to implement some of your suggestions in the limited time we had. Specifically, we now added 2 new visualizations to the revised appendix (Figs. S17 & S18 in the Appendix). As suggested, we now applied t-SNE to the embeddings of all 8 subjects. We supply 2 visualizations of that t-SNE:
>
> - Fig. S16 shows 2D t-SNE visualization of brain streams (e.g. early, lateral and parietal) of all brain voxels (of all 8 subjects). It shows that the different brain streams (marked by different point colors) are well separated in the voxel-embedding space, whereas different subjects (marked by different point shapes) are intermingled.
> - Fig. S17 shows 2D t-SNE visualization of 4 specific category selective regions (faces, bodies, places, and words)  – the colors now signify their pre-defined cortical regions, whereas the proximity is imposed by the similarity of our voxel-embeddings. As can be seen, our voxel embedding implicitly learned functionality which is similar to the functionality of known category selective brain regions.
>
>
> **Q2.** ***In other papers that use NSD, subject 5 is typically the best performing subject. However in this work the retrieval accuracy is quite a bit lower than subjects 1 and 2. Any ideas why this might be the case?***
>
> **Response:** This is an interesting observation. In our analysis, Subject5 from NSD does indeed perform best in terms of median Pearson correlation, as shown in Figure 4a in the paper. However, as the reviewer noted, this does not hold for retrieval accuracy in Figure 4b. One possible explanation could be that the metrics used for assessing fMRI prediction quality and fMRI retrieval are different:  For assessing the quality of fMRI prediction, we used the standard method of per-voxel Pearson correlation across all the test images (~1000). However, this measure cannot be used for image retrieval. We therefore used for retrieval the similarity between 2 fMRIs (the predicted fMRI and the ground-truth fMRI).

---

### Official Review · Reviewer_Z1zy · 2024-11-06

**Soundness:** 2
**Presentation:** 3
**Contribution:** 2
**Rating:** 3
**Confidence:** 5

**Summary:**

This paper introduces a Universal fMRI Encoder for the prediction of brain responses to image stimuli. Unlike traditional subject-specific brain encoding models, the proposed work is trained and validated across multiple subjects and datasets. The model learns voxel embedding through cross-attention with multi-level deep image features, allowing the model to capture functional roles of different brain regions while sharing other network weights across subjects. The model is evaluated on 3 datasets on two measurements: a comparison of the estimated fMRI signal vs. ground truth and the image retrieval accuracy using top-k accuracy.

**Strengths:**

1. The proposed Universal Brain-Encoder can effectively handling sequences from different subjects, datasets, and machines, which enhances its applicability for both neuroscience research and practical applications

2. The paper presents comprehensive experimental results, and the proposed Universal Brain-Encoder achieves satisfied performance across multiple datasets. Notably, it achieves substantial performance improvements when trained on multi-dataset inputs, supporting the authors' argument regarding the "Crowd of Brains" concept

**Weaknesses:**

1. The idea appears to closely resemble existing works such as [1], MindFormer [2], MindEye2 [3], MindBridge [4], and BDI [5]. These studies also learn a set of independent parameters for each subject while sharing most parameters across subjects. The novelty of the proposed idea needs further clarification.

2. Some brain decoding methods employ symmetric architectures, so they have both Image-to-fMRI and fMRI-to-Image networks, such as [6] and [7]. A discussion about these approaches should be included in the comparative experiments.

3. The quality of the generated fMRI data requires more validation. The authors should use additional metrics or evaluation methods to assess whether the generated data can still be used to analyze brain activity. For instance, the authors can use existing brain decoding models to prove they can reconstruct images from the generated fMRI sequences.

4. An ablation study on different Voxel Embedding dimensions should be included.

5. Is it better to utilize all voxels in fMRI sequences? The proposed voxel-based approach has the potential to capture latent semantic relationships between brain activities and input signals, whereas manually selected ROIs may lead to information loss. If the method can effectively model all voxels and provide visualize results as demonstrated in Fig. 7, it would yield interesting results.

[1] Functional Brain-to-Brain Transformation with No Shared Data
[2] MindFormer: A Transformer Architecture for Multi-Subject Brain Decoding via fMRI
[3] MindEye2: Shared-Subject Models Enable fMRI-To-Image With 1 Hour of Data
[4] MindBridge: A Cross-Subject Brain Decoding Framework
[5] Brain Dialogue Interface (BDI): A User-Friendly  fMRI Model for Interactive Brain Decoding
[6] From voxels to pixels and back: Self-supervision in natural-image reconstruction from fMRI
[7] Rethinking Visual Reconstruction: Experience-Based Content Completion Guided by Visual Cues

**Questions:**

What is the spatial resolution of the pre-processed fMRI datasets and the corresponding dimensionality of the 4D volumetric data? It is curious whether the spatial resolution can support the fine-grained analysis of brain response as shown in Fig. S13.

**Details Of Ethics Concerns:**

No ethics concerns are found in this work, and all datasets used are from the public domain.

---

> ### Author Response · Authors · 2024-11-20
>
> Thank you for your time and comments. Below are our answers to your questions and suggested experiments. Please let us know if this resolves your concerns, or whether there are additional comments/requests.
>
> **Q1.** ***The idea appears to closely resemble existing works such as [1], MindFormer [2], MindEye2 [3], MindBridge [4], and BDI [5]. These studies also learn a set of independent parameters for each subject while sharing most parameters across subjects. The novelty of the proposed idea needs further clarification.***
>
> **Response:** Thanks for pointing out these papers. We are familiar with most of them (and will make sure to refer to them in the paper). We would like to point out what distinguishes our approach from prior work. These prior papers are all fMRI-centric. Namely, they treat each fMRI scan as a single entity, relying on shared embedding for the ENTIRE fMRI across subjects. In contrast, our model is VOXEL-centric, hence searches for shared VOXEL-embeddings across subjects. Our approach shares network weights across all brain-voxels (regardless of subject, dataset, or machine), with only the voxel-specific embeddings being unique. Similar voxel behaviour is more likely to exist, than similarity of entire fMRI response across different brains. This fundamental difference is what allows:
> (1) Effective integration of data across different subjects/datasets/machines.
> (2) Simple transfer learning to new brains/datasets with few data samples.
> (3) Facilitates exploration of brain regions/functionality.
>
> Moreover, please note that most of the above-mentioned papers do not address the problem of Image-to-fMRI encoding (which is the focus of our paper). We will clarify all these points. However, we agree that these papers are closely related, and will refer to them properly in our Prior Work section.
>
>
> **Q2.** ***Some brain decoding methods employ symmetric architectures, so they have both Image-to-fMRI and fMRI-to-Image networks, such as [6] and [7]. A discussion about these approaches should be included in the comparative experiments.***
>
> **Response:** These methods which employ symmetric architectures, primarily aim for fMRI-to-Image reconstruction rather than Image-to-fMRI encoding. Note that in our paper, we do provide comparisons with the encoder from Gaziv et al., 2022 (which is a later  improved version of [6] by the same above). We further include comparisons with Takagi & Nishimoto, 2023 (see appendix). Both these methods utilize an encoder-decoder framework. As can be seen in our paper (see Fig. 4 and Appendix Fig. S8), our encoder significantly outperforms these methods.
>
> **Q3.** ***The quality of the generated fMRI data requires more validation. The authors should use additional metrics or evaluation methods to assess whether the generated data can still be used to analyze brain activity. For instance, the authors can use existing brain decoding models to prove they can reconstruct images from the generated fMRI sequences.***
>
> **Response:** Following the reviewer’s request, we have now also conducted an Image Reconstruction experiment to validate the quality of our predicted fMRI. For this experiment, we used one of the existing brain decoding models (MindEye). Our new experimental results show that image reconstruction from our predicted fMRI is on par with image reconstruction from ground-truth fMRI. Please see Fig. S13 and Table S5 in the revised Appendix (showing both quantitative and qualitative results).
>
> In the paper we previously assessed the quality of the encoded fMRI data using the widely accepted metric of mean Pearson correlation, as done in most previous studies. We also conducted Retrieval experiments (Figs. 3, 4) to further validate the quality of our predicted fMRI. Altogether, these 3 types of evaluations validate our encoder's high performance, demonstrating its effectiveness beyond the standard methods used in prior work.
>
>
> **Q4.** ***An ablation study on different Voxel Embedding dimensions should be included.***
>
> **Response:** Following the reviewer’s suggestion, we performed an ablation study on different voxel embedding dimensions E. We experimented with E= 64, 128, 256, 512, 1024. The ablation evaluated on Subject1 is as follows: Median Pearson Correlation = 0.5170, 0.5221, 0.5221, 0.5227, 0.5243.  Our experiments show that the results are not very sensitive to the size of E. We added this ablation to the revised appendix (Table S4).

---

> ### Author Response · Authors · 2024-11-20
>
> **Q5.** ***Is it better to utilise all voxels in fMRI sequences? The proposed voxel-based approach has the potential to capture latent semantic relationships between brain activities and input signals, whereas manually selected ROIs may lead to information loss. If the method can effectively model all voxels and provide visualised results as demonstrated in Fig. 7, it would yield interesting results.***
>
> **Response:** In our study, we used all brain voxels in the entire visual cortex, and additionally  voxels in other brain regions which are considered to be related to visual processing. Since our method is voxel-centric, it is NOT limited to a particular choice of any ROI, and can be applied in principle to any brain voxel. However, the prediction of the voxel activity of any brain voxel will be good only if that voxel is sensitive to visual data. Our functional-clustering results of all brain voxels we used in this study can be found in Fig. S14 in the Appendix.
>
> **Q6.** ***What is the spatial resolution of the pre-processed fMRI datasets and the corresponding dimensionality of the 4D volumetric data? It is curious whether the spatial resolution can support the fine-grained analysis of brain response as shown in Fig. S13.***
>
> **Response:** The spatial resolution of the fMRI recordings is 1.8 mm isotropic, with a temporal sampling rate of 1.6 seconds. The brain map in Figure S16 (was S13) visualises data at the voxel level, so the spatial resolution corresponds directly to that of the fMRI resolution. Given that NSD is a high-resolution 7-Tesla dataset, it supports fine-grained analysis of brain responses, as demonstrated in that figure.

---

### Public Comment · ~Zhibo_Tian1 · 2024-11-13
**Mapping Voxel Embeddings to Brain Regions.**

Dear Authors,

I have a question regarding the mapping of voxel embeddings to brain regions. Given that each voxel embedding is a 256-dimensional vector, could you please clarify how these embeddings are associated with specific brain regions, particularly in Section 5, "Exploring the Brain Using Voxel-Embeddings"?

Thank you.

---

> ### Author Response · Authors · 2024-11-20
>
> Thank you for your interest in our work. Each learned voxel embedding corresponds to a specific voxel (for which we know its 3D location in the brain). In Fig. 7a, we use K-means clustering on the learned voxel embeddings of 2 subjects, grouping embeddings into clusters. Each cluster represents a set of voxel embeddings and their corresponding brain voxels, which we refer to as a discovered functional brain region. To explore the functionality of each region, we visualize the images with the highest activation for that cluster by calculating the mean voxel activation (of voxels in the cluster) across all images. For fine-grained functionality (Fig. 7b), we apply the same procedure to voxel embeddings whose corresponding voxels belong to a predefined functional region (ROI).

---

### Author Response · Authors · 2024-11-20

We would like to thank all the reviewers for their time, effort and valuable comments. We did our best to address all the concerns and the extra requested evaluations. The new results, evaluations and experiments we performed per the reviewers’ suggestions are now added to the Revised Appendix of the uploaded version.

Please note that the main paper itself has not been revised yet (due to the time limit). Pointers to these new experiments from the main paper will be added in the final version of the paper, as well as the added clarifications below.

---

### Author Response · Authors · 2024-11-28

Thank you for all your time, valuable comments, and suggestions. We will incorporate all the new experimental results we reported in our rebuttal (per the reviewers‘ requests), as well as the missing citations and our textual clarifications (following the discussion with the reviewers), into the final version of our paper.  This will undoubtedly make our paper clearer to future readers. 😊

We will add all these new additions either in the body of the paper or in the appendix (pending on the page-limit constraints of the main body). If added to the appendix, we will add a clear and proper reference to its location in the appendix from the paper itself.


Thank you for all your feedback!

---

### Meta-Review · Area_Chair_AYPC · 2024-12-23

**Metareview:**

This manuscript proposes a model for the prediction of volumes of fMRI (BOLD) activations given images, presumably image stimuli. It claims generalization ("universal" encoder) allowing for predictions to be made in a variety of settings in a variety of subjects. The authors demonstrate use empirically on three open datasets (`vim-1`, `fMRI-on-ImageNet`, and `NSD`).

Overall the concept of encoding and decoding for brain signals (of a variety of modalities) is not novel, and the authors accordingly provide a bevy of references. The reviewers contend that additional references may be considered as baselines or related, and the authors do not compare to them; rebuttal by the authors states that 1) the models given are mostly full brain encoders instead of voxel/vertex-wise, and 2) the listed are mostly decoder models. The authors instead compare only to the encoder branch of Gaziv et al. 2022 (NeuroImage), which is listed as "Baseline".

It seems unclear to me what the contribution of this particular manuscript provides over, e.g., the similar Gaziv et al. 2022 paper. I tend to agree with reviewer `Z1zy`, this manuscript appears to resemble previous work in form, and only provides parameterization to inter-subject variation. Moreover, it appears to be a straightforward application of transfer learning with respect to the population variation.

It occurs to me that in reading this paper, I feel that I have not learned anything extraordinary since reading the Gaziv et al. 2022 paper nor its immediate predecessor Beliy et al. 2019. This is not to say there haven't been improvements, or that the individual works aren't themselves of value, but that in re-reading all three in immediate succession during the review of `Z1zy`'s comments and its replies by the authors, I find low marginal information gain on the later papers conditional on the earlier publications. It seems that (deep) fMRI-Image decoder/encoder pairings are becoming well explored, and I think that producing work that either educates us further about the brain or about models themselves is required for truly outstanding work.

I therefore recommend rejection, but I note to the SACs/PC Chairs that this paper is quite borderline, and I am happy to be overruled. The manuscript is in acceptable form, but appears as a derivative work from transfer learning on previous methods.

**Additional Comments On Reviewer Discussion:**

I apologize to the authors for the scant discussion of the other reviews, and for their somewhat ad hoc notes.

---

### Decision · Program_Chairs · 2025-01-22

Reject